# Nucleoside-modified VEGFC mRNA induces organ-specific lymphatic growth and reverses experimental lymphedema

Dániel Szőke [1,2], Gábor Kovács [1,2], Éva Kemecsei[1,2], László Bálint [1,2], Kitti Szoták-Ajtay[1,2], Petra Aradi[1,2], Andrea Styevkóné Dinnyés[1,2], Barbara L. Mui[3], Ying K. Tam[3], Thomas D. Madden[3], Katalin Karikó [4], Raghu P. Kataru[5], Michael J. Hope[3], Drew Weissman[6], Babak J. Mehrara[5], Norbert Pardi[6✉] & Zoltán Jakus [1,2✉]

Lack or dysfunction of the lymphatics leads to secondary lymphedema formation that seriously reduces the function of the affected organs and results in degradation of quality of life. Currently, there is no definitive treatment option for lymphedema. Here, we utilized nucleoside-modified mRNA encapsulated in lipid nanoparticles (LNPs) encoding murine Vascular Endothelial Growth Factor C (VEGFC) to stimulate lymphatic growth and function and reduce experimental lymphedema in mouse models. We demonstrated that administration of a single low-dose of VEGFC mRNA-LNPs induced durable, organ-specific lymphatic growth and formation of a functional lymphatic network. Importantly, VEGFC mRNA-LNP treatment reversed experimental lymphedema by restoring lymphatic function without inducing any obvious adverse events. Collectively, we present a novel application of the nucleoside-modified mRNA-LNP platform, describe a model for identifying the organ-specific physiological and pathophysiological roles of the lymphatics, and propose an efficient and safe treatment option that may serve as a novel therapeutic tool to reduce lymphedema.

[1] Department of Physiology, Semmelweis University School of Medicine, Budapest, Hungary. [2] MTA-SE „Lendület" Lymphatic Physiology Research Group of the Hungarian Academy of Sciences and the Semmelweis University, Budapest, Hungary. [3] Acuitas Therapeutics, Vancouver, BC, Canada. [4] BioNTech RNA Pharmaceuticals, Mainz, Germany. [5] Department of Surgery, Division of Plastic and Reconstructive Surgery, Memorial Sloan Kettering Cancer Center, New York, NY, USA. [6] University of Pennsylvania, Perelman School of Medicine, Philadelphia, PA, USA. ✉email: pnorbert@pennmedicine.upenn.edu; jakus.zoltan@med.semmelweis-univ.hu

The lymphatic system regulates interstitial fluid balance, immune cell trafficking, and lipid absorption from the small intestine[1,2]. Besides these well-known, classical functions, several novel and unexpected physiological and pathophysiological functions of the lymphatics have been recently revealed including blood pressure regulation, reverse cholesterol transport, lymphatic function in the meningeal compartment and an important role in the preparation for neonatal respiration[1–10]. The damage of lymphatic vessels due to tumor surgery, trauma, or infection leads to secondary lymphedema development that seriously reduces the function of the affected organs and results in the degradation of quality of life[1,2,11,12]. Development of secondary lymphedema after axillary lymphadenectomy in breast cancer patients is the most common non-infectious cause of the disease[1,11–13]. In addition to swelling due to the accumulation of extracellular fluid, lymphedema may result in recurrent local bacterial and fungal infections and fibrosis[1,11,12]. Currently used therapeutic approaches such as manual lymph drainage and compression therapy have great limitations and provide only symptomatic relief[14]. Thus, there is a great need for an effective treatment option to reduce the symptoms or cure this disease.

Vascular Endothelial Growth Factor C (VEGFC) is the most important lymphangiogenic factor, which, as a ligand for Vascular Endothelial Growth Factor Receptor 3 (VEGFR3), induces VEGFR3-dependent lymphatic growth[15–17]. Loss of VEGFC-VEGFR3 signaling leads to blockade of lymphatic development, growth, and regeneration[15]. Therefore, VEGFC is an important therapeutic target to induce lymphatic regeneration in lymphedema.

Protein therapeutics is a rapidly growing segment of the pharmaceutical industry[18]. However, the use of proteins has major drawbacks, such as high cost of manufacturing, the need for specific and often complicated purification procedures, and difficulties maintaining effective levels of proteins with short biological half-lives. The use of recombinant VEGFC to induce lymphatic regeneration in vivo has been evaluated, but these studies revealed limited effect in in vivo models[5,19–22]. For example, Hall and colleagues revealed that VEGFC protein treatment induced an increase in the number of lymphatic vessel branching points and junctions but these changes were not significant during the process of wound healing[19]. Szuba and coworkers showed that high doses of the lymphangiogenic factor was needed (100 μg of recombinant VEGFC injected into the rabbit ear) to induce the proliferation of lymphatic vessels[20]. To address these limitations, albumin-alginate microparticle or hydrogel bound forms of VEGFC protein were used to regulate the rate of protein release and extend the in vivo effect in murine heart after myocardial infarction or meningeal compartment[5,21]. Da Mesquita and colleagues demonstrated that it was not possible to detect the growth of meningeal lymphatic structures after recombinant VEGFC treatment, only slight dilation of the vessels was apparent[5].

To find other ways to address the limitations of protein therapeutics, adenovirus and adeno-associated virus (AAV)-based platforms appear to be promising alternatives. Adenovirus and AAV-based delivery of human and murine VEGFC has been shown to be effective in mice and domestic pigs to induce lymphatic growth; thus, a drug candidate entered a Phase II clinical trial (NCT03658967)[1,23–31]. Importantly, these delivery platforms have major drawbacks such as pre-existing host immunity, anti-vector immune responses, difficulties with regulation and toxicity that may hinder application in clinical practice[32]. For example, genomic integration of AAV2 has been shown to lead to hepatocellular carcinoma formation that raises serious safety concerns about this delivery approach[33].

In vitro transcribed messenger RNA (mRNA) emerged as a potent platform for infectious disease and cancer vaccine development (reviewed in[34]), protein replacement therapy[35–37], and in vivo genome editing[38]. mRNA-based platforms have several important advantages over other types of therapeutic protein delivery systems, such as the lack of integration into the host genome, no anti-vector immunity after repeated administrations, highly controllable transient protein production, and rapid, scalable, and sequence-independent manufacturing that does not require complex and expensive infrastructure[34]. One of the most promising mRNA-based therapeutic platforms is nucleoside-modified mRNA encapsulated in lipid nanoparticles[39] (LNPs) that has demonstrated safety and efficacy for a wide range of applications in various preclinical models[34–38]. Most importantly, this leading technology is currently being evaluated in Phase III clinical studies for SARS-CoV-2 vaccine development (NCT04470427, NCT04368728) and it has received approval for mass vaccination in multiple countries around the globe[40,41].

Herein, nucleoside-modified mRNA-LNPs encoding murine VEGFC were utilized to stimulate lymphatic growth and function and reduce experimental lymphedema in mice. These studies revealed that the administration of a single low-dose of VEGFC mRNA-LNPs induces organ-specific lymphatic growth and active lymphatic function in the newly formed lymphatic vessels in mice. Importantly, the VEGFC mRNA-LNP treatment reverses secondary experimental lymphedema without showing any obvious adverse events in an animal model.

## Results

### VEGFC mRNA-LNPs induce VEGFC secretion in vitro and local protein production in vivo.
To develop a novel platform to modulate lymphatic growth, murine VEGFC-encoding 1-methylpseudouridine-containing mRNAs were designed, synthetized, purified, and encapsulated into LNPs. As a first characterization of the modified mRNA-LNP platform, HEK293T cells were treated with 1 μg of Poly(C) RNA-LNPs (negative control) or GFP mRNA-LNPs, and GFP expression was detected by Western blot analyses and fluorescent microscopy at various time points. As it is shown in Supplementary Fig. 1, GFP mRNA-LNPs triggered expression of high levels of GFP protein in HEK293T cells.

HEK293T cell cultures were then transfected with 1 μg of Poly (C) RNA-LNPs or VEGFC mRNA-LNPs and cell lysates and cell culture supernatants were subjected to Western blot and ELISA analyses 8 hours, 1, 2, 4, 8, and 12 days after the treatment. Production of high levels of VEGFC protein demonstrated the ability of the mRNA-LNP construct to induce protein expression in vitro (Supplementary Fig. 2a–c).

Next, a series of studies was conducted to investigate the effects of VEGFC mRNA-LNP administration in vivo. When 1 μg of VEGFC mRNA-LNP was injected into the ear of mice, higher level of VEGFC protein was detectable in the interstitial fluid of the ear compared to the Poly(C) RNA-LNP-injected animals 1, 5, 10, 15, and 20 days after treatment (Fig. 1a). VEGFC protein expression peaked 1 day after the treatment and VEGFC expression was significantly higher compared to the control until 15 days post VEGFC mRNA-LNP injection.

### VEGFC mRNA-LNP treatment results in local lymphatic growth in vivo.
Thereafter, the effect of VEGFC mRNA-LNP treatment on lymphatic growth was tested in *Prox1*[GFP] lymphatic reporter mice, in which lymphatic endothelial cells express GFP[42]. Strikingly, a massive increase in lymphatic growth was found after 22 and even after 60 days following intradermal injection of as little as 1 μg of VEGFC mRNA-LNPs into the ear shown by fluorescent stereo microscopy and detecting the Prox1-GFP signal and LYVE1 expression of lymphatic endothelial cells in paraffin-embedded histology slides (Fig. 2a–b).

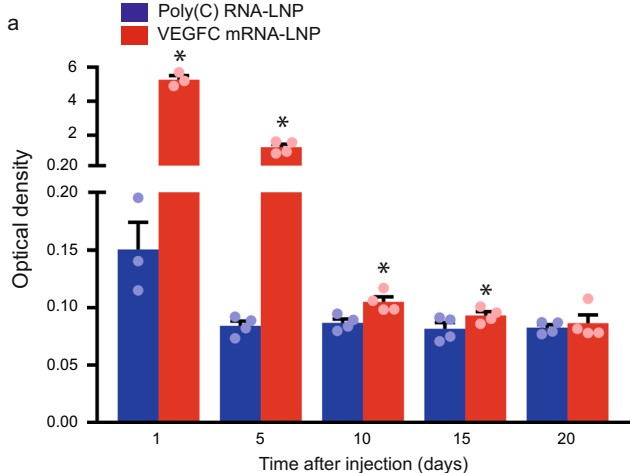

**Fig. 1 Administration of VEGFC mRNA-LNPs induces VEGFC secretion in vivo. a** 1 µg of Poly(C) or VEGFC mRNA-LNPs were injected into contralateral ears of mice intradermally. Interstitial fluid samples were harvested 1, 5, 10, 15, and 20 days post injection and VEGFC protein expression levels were determined by ELISA. Quantitative data for secreted amount of VEGFC are represented as mean and SEM from Poly(C) or VEGFC mRNA-LNP injected ears of 3–4 mice at each time point (two-tailed, paired $T$-test, $P = 0.0023$ after 1 day for 3 mice, $P = 0.0063$ after 5 days for 4 mice, $P = 0.0017$ after 10 days for 4 mice, $P = 0.0332$ after 15 days for 4 mice, and $P = 0.7025$ after 20 days for 4 mice). Asterisks indicate $P < 0.05$ compared with control.

Next, lymphatic growth after VEGFC mRNA-LNP injection was quantified by measuring the length of the lymphatic network, the average diameter of the lymphatic vessels, and by counting the number of the branching points. We measured significant increases in all three parameters, and importantly, the effect of VEGFC mRNA-LNP treatment was maintained for an extended period of time (up to 60 days) both in the ear (Fig. 2c) and back skin (Supplementary Fig. 3a, b). Of note, contralateral injections were performed into the right and left sides of the same animal in these experiments. Next, a dose response study was performed by injecting animals with 0.04, 0.2, 1, and 5 µg of VEGFC mRNA-LNPs. Significantly increased lymphatic growth was observed in mice injected with higher doses of VEGFC mRNA-LNPs in the back skin (Supplementary Fig. 3c) and ear (Fig. 2d and Supplementary Fig. 3d). The effect of 1 µg of VEGFC mRNA-LNP treatment of the ear was also quantified by flow cytometry in *Prox1*[GFP] mice. As it is shown in Fig. 2e, VEGFC mRNA-LNP injection resulted in an increase of the number of GFP-positive cells, confirming our previous results.

In comparison to the nucleoside-modified mRNA-LNP system, we have assessed the effect of recombinant human VEGFC protein treatment in vivo (Supplementary Fig. 4). Intradermal injection of 1 µg of recombinant VEGFC protein into the ear and back skin induced modest lymphatic growth after the treatment detected by stereo microscopy in lymphatic reporter mice, suggesting insufficient duration of activity and limited effect of the injected protein (Supplementary Fig. 4).

To further demonstrate the versatility of our VEGFC mRNA-LNP-based approach, observations were extended to other organs. Increased lymphatic growth in the diaphragm after intraperitoneal injection, in the lungs after intratracheal injection, and in the musculus gastrocnemius after intramuscular administration of VEGFC mRNA-LNP was detected, suggesting that the

nucleoside-modified VEGFC mRNA-LNP platform is an efficient tool to induce lymphatic growth in various organs after in vivo administration (Fig. 2f and Supplementary Fig. 5a).

Proliferation of the lymphatic endothelial cells was monitored by determining the number of 5-ethynyl-2′-deoxyuridine (EdU) positive lymphatic endothelial cells and calculating the mitotic index 24 h post EdU administration. Both the number and fraction of EdU positive nuclei of lymphatic endothelial cells were highly increased in ears injected with 1 µg of VEGFC mRNA-LNP compared to the control shown by fluorescent and confocal microscopy (Fig. 3a–c).

**VEGFC mRNA-LNP treatment induces organ-specific lymphatic growth after intradermal delivery.** Previous studies revealed local production of mRNA-LNP-encoded protein after intradermal delivery[39]. To confirm these findings, C57Bl/6 mice were intradermally injected with 1 µg of GFP mRNA-LNPs in the back skin (or the ear in separate animals), and GFP expression was monitored in the injected back skin or ear, lung, and small intestine. While strong local GFP signal was present at the sites of injections in the back skin or ear and there was no detectable GFP protein in the lung and small intestine of the same animal (Supplementary Fig. 6).

To further confirm local protein production and rule out potential off-target effects after intradermal mRNA-LNP delivery, 1 µg or 5 µg of VEGFC mRNA-LNP was administrated to the back skin (or in a separate experiment to the ear) and lymphatic growth was monitored in the ears, back skin, lung, and small intestine of the same animal by fluorescent microscopy and flow cytometry 21 days after the treatment (Fig. 4a–c). VEGFC-induced lymphangiogenesis was limited to the injection sites (back skin or injected ear) and no lymphatic growth was detectable in other organs including the other ear (non-injected), lung, and small intestine shown by fluorescent microscopy (Fig. 4a–b) and flow cytometry (Fig. 4c). These results indicate that the nucleoside-modified VEGFC mRNA-LNP platform induces local VEGFC secretion and organ-specific lymphatic growth in the ear or back skin after intradermal delivery.

**VEGFC mRNA-LNP administration results in non-significant blood vessel proliferation and immune response activation compared to delivery of Poly(C) RNA-LNP.** It is well known that adenovirus and AAV-based delivery of genes induces robust activation of the immune system, and may have an impact on blood vessel proliferation[32,43]. In contrast, a recent study demonstrated that systemic administration of nucleoside-modified mRNA-LNPs did not induce inflammatory responses[44]. To further demonstrate target specificity and safety of nucleoside-modified VEGFC mRNA-LNPs, a series of additional studies were performed.

First, it was demonstrated that intradermal treatment of the mouse ear or back skin with VEGFC mRNA-LNPs did not result in local blood vessel proliferation compared to Poly(C) RNA-LNP treatment on day 22 post injection shown by high von Willebrand Factor (vWF) blood vessel marker expression, the pan-endothelial marker CD31 staining and the absence of LYVE1 signal, while the VEGFC mRNA-LNP-induced lymphatic growth was present shown by LYVE1 and Podoplanin immunostaining of lymphatic endothelial cells (Fig. 5a–b and Supplementary Fig. 7a–b).

Second, the immune response induced by intradermal VEGFC mRNA-LNP treatment compared to delivery of Poly(C) RNA-LNP was monitored 22 days post injection. While VEGFC mRNA-LNP treatment resulted in an increase in the number of the LYVE1 positive lymphatic endothelial cells shown by flow

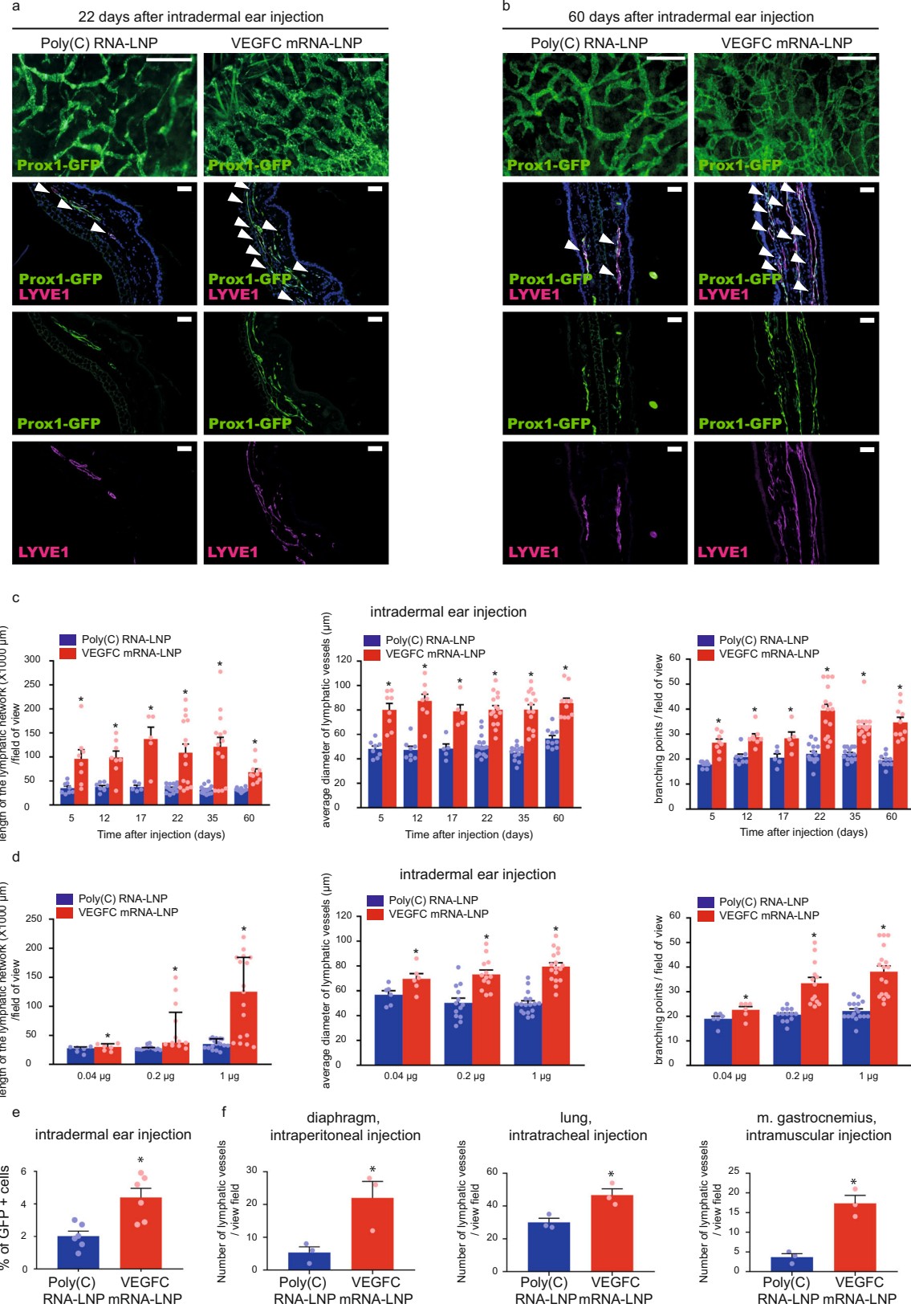

cytometry, it did not induce a significant change in the number of CD45 positive immune cells, Ly6G/C (GR1) positive granulocytes, CD206 positive cells including macrophages, CD3 positive T lymphocytes, B220 positive B lymphocytes and Ly6G/C (GR1) negative and CD11b positive monocytes compared to Poly(C) RNA-LNP treatment (Fig. 5c). In addition, no significant Ly6G/C (GR1) positive immune cell infiltration was observed in paraffin-embedded tissues (Fig. 5d).

**Fig. 2 Administration of VEGFC mRNA-LNPs induces local lymphatic growth in mice in vivo. a–b** Analysis of lymphatic morphology in the ear of *Prox1*[GFP] lymphatic reporter mice injected with 1 μg of Poly(C) or VEGFC mRNA-LNPs. Representative images 22 days (**a**) and 60 days (**b**) after the ear treatment of 15 (**a**) and 5 (**b**) mice in each group are shown by whole-mount fluorescent stereo microscopy (upper panels; bars, 1000 μm (**a**) and 500 μm (**b**)) and Prox1-GFP signal and LYVE1 immunostaining of slides processed by paraffin-based histology are shown (lower panels; bars, 50 μm). Arrows indicate Prox1-GFP and LYVE1 double positive lymphatic vessels. **c** Assessment of the time-dependent effect of intradermal administration of 1 μg of Poly(C) or VEGFC mRNA-LNPs in the ear at days 5, 12, 17, 22, 35, and 60. Quantitative data for the length of lymphatic network, average lymphatic vessel diameter, and number of branching points per field of view are represented as mean and SEM from Poly(C) or VEGFC mRNA-LNP-injected ears of 5–15 mice in each group (two-tailed, paired *T*-test, for lymphatic network length $P = 0.0005$ after 22 days for 15 mice and $P = 0.0005$ after 60 days for 10 mice; for average lymphatic vessel diameter $P = 0.000005$ after 22 days for 15 mice and $P = 0.0002$ after 60 days for 10 mice; for number of branching points $P = 7.48 \times 10^{-7}$ after 22 days for 15 mice and $P = 0.0001$ after 60 days for 10 mice). **d** Monitoring the dose-dependent effect of Poly(C) or VEGFC mRNA-LNPs (0.04, 0.2, and 1 μg) 20 days after intradermal treatment of the ear. Quantitative data for the length of lymphatic network, average lymphatic vessel diameter, and number of branching points per field of view are represented as mean and SEM from LNP complex-injected ear of 6–17 mice in each group. (two-tailed Wilcoxon signed-rank test for lymphatic network length $P = 7.04 \times 10^{-5}$ for 17 mice, two-tailed, paired *T*-test for average lymphatic vessel diameter $P = 4.61 \times 10^{-7}$ for 17 mice and two-tailed, paired *T*-test for number of branching points $P = 6.84 \times 10^{-7}$ for 17 mice when injected with 1 μg of Poly(C) or VEGFC mRNA-LNP). **e** Monitoring the effect of intradermal injection into ears of *Prox1*[GFP] mice with 1 μg of Poly(C) or VEGFC mRNA-LNPs shown by flow cytometry analysis. Quantitative data for GFP positive cell number are represented as mean and SEM from Poly(C) or VEGFC mRNA-LNP-injected ears of 6 mice in each group, 22 days after mRNA-LNP injection (two-tailed, paired *T*-test, $P = 0.0010$ for 6 mice). **f** Monitoring the effect 1 μg of locally injected Poly(C) or VEGFC mRNA-LNPs on lymphatic growth in *Prox1*[GFP] mice in the diaphragm 22 days after intraperitoneal injection, in the lungs 22 days after intratracheal treatment, and in the musculus gastrocnemius 22 days after intramuscular injection. Quantitative data for number of the lymphatics are shown as mean and SEM of Poly(C) or VEGFC mRNA-LNP-injected organs of 3 mice in each group. Asterisks indicate $P < 0.05$ compared with control. (two-tailed, unpaired *T*-test, diaphragm: $P = 0.0354$ for 3 mice, lung: $P = 0.0222$ for 3 mice, two-tailed, paired *T*-test m. gastrocnemius: $P = 0.0145$ for 3 mice).

**VEGFC mRNA-LNP treatment results in formation of fully functional lymphatic vessels in mice.** Our results indicate that the nucleoside-modified VEGFC mRNA-LNP platform is an efficient and safe tool to induce VEGFC secretion and organ-specific lymphatic growth in vivo. Although the morphology of the newly formed lymphatic vessels appeared to be normal, additional studies were performed to confirm the functionality of these lymphatic vessels. It is well-documented that large molecular weight Rhodamine dextran (Rh-D) is taken up and transported by the lymphatic vessels, therefore, it is an excellent tool to monitor lymphatic function in vivo[4,9,10]. To this end, *Prox1*[GFP] mouse ears after 22 days and hind paws after 75 days of intradermal treatment with Poly(C) RNA-LNP or VEGFC mRNA-LNP were injected with 70 kDa Rh-D. Our results indicate that lymphatic vessels take up and transport the large molecular weight molecules after control Poly(C) RNA-LNP treatment in both the ears and hind limbs (Fig. 6a–b), which demonstrates that this method is an appropriate tool to monitor lymphatic function in the ear and hind limb. Importantly, increased number of lymphatic vessels can be detected in the VEGFC mRNA-LNP-injected organs, and this dense network of lymphatic vessels also takes up and transports the 70 kDa Rh-D without signs of significant leakage of the macromolecule after 22 days in the ear and after 75 days in the hind paw following the treatment (Fig. 6a–b). The transport of the labeled macromolecule to the popliteal lymph node can also be detected (Fig. 6b). These results indicate that VEGFC mRNA-LNP treatment not only induces the proliferation of lymphatic vessels but these new vessels are also functional even after an extended period of time.

These studies raised the possibility that the VEGFC mRNA-LNP treatment induces the proliferation of not only the lymphatic capillaries but also collecting lymphatic structures. In connection, an increase in the number of the lymphatic vessels with smooth muscle coverage in the VEGFC mRNA-LNP-treated samples was detected (Fig. 6c) suggesting that VEGFC mRNA-LNP treatment may induce the proliferation of collecting lymphatics. Many of the newly formed structures with smooth muscle coverage appeared to be EdU positive after VEGFC mRNA-LNP treatment also suggesting the formation of new collecting lymphatic vessels (Fig. 6d–e). Collectively, the new lymphatic vessels—induced by the VEGFC mRNA-LNP

treatment—are fully functional, and the new platform may also stimulate the growth of collecting lymphatics.

**Disease induction results in secondary lymphedema development in a genetic mouse model.** To provide proof-of-concept and test whether nucleoside-modified VEGFC mRNA-LNP treatment influences disease progression in an experimental lymphedema system, we assessed the present platform in a recently developed Diphtheria toxin-inducible genetic secondary lymphedema mouse model[45,46]. VEGFR3 (encoded by the *Flt4* gene) expression is restricted to lymphatic endothelial cells in adults allowing efficient targeting of lymphatic endothelial cells[17,47]. Expression of the Diphtheria toxin receptor (DTR) in lymphatic endothelial cells was induced by repeated intraperitoneal tamoxifen injections into Flt4-CreER[T2]; *iDTR*[fl/fl] mice. Thereafter, repeated intradermal injections of PBS or Diphtheria toxin were performed into the hind paw of the tamoxifen-treated Flt4-CreER[T2]; *iDTR*[fl/fl] mice. Of note, the PBS and Diphtheria toxin were injected into the contralateral paw of the same animal. Eight days after Diphtheria toxin treatment, significant swelling of the paw was detected. Assessment of the clinical signs of lymphedema revealed an increase in clinical score, as well as paw thickness in the Diphtheria toxin-injected group after treatment initiation (Fig. 7a). Staining of histology slides with Haematoxylin and Eosin also confirmed the development of secondary lymphedema in this model (Fig. 7b). Our data also demonstrates the increase of the fibroadipose area and Collagen I deposition in Diphtheria toxin-treated Flt4-CreER[T2]; *iDTR*[fl/fl] mice and indicate the efficient elimination of lymphatic endothelial cells in the model (Fig. 7c–f). Next, we injected 70 kDa Rh-D into the hind paw in equal amount in each group, and monitored the lymphatic function 90 min after injection of the tracer. Importantly, the drainage of labeled macromolecules to the popliteal lymph node was blocked in Flt4-CreER[T2]; *iDTR*[fl/fl] mice 30 and 75 days after the Diphtheria toxin treatment (Fig. 8a–b). Of note, some lymphatic structures can be detected in the Diphtheria toxin treated group, but these lymphatic endothelial cells do not form a functional network as it was also described earlier[46] (Figs. 7d–e and 8a–b). In conclusion, the experimental mouse model of secondary lymphedema induction has been rigorously validated.

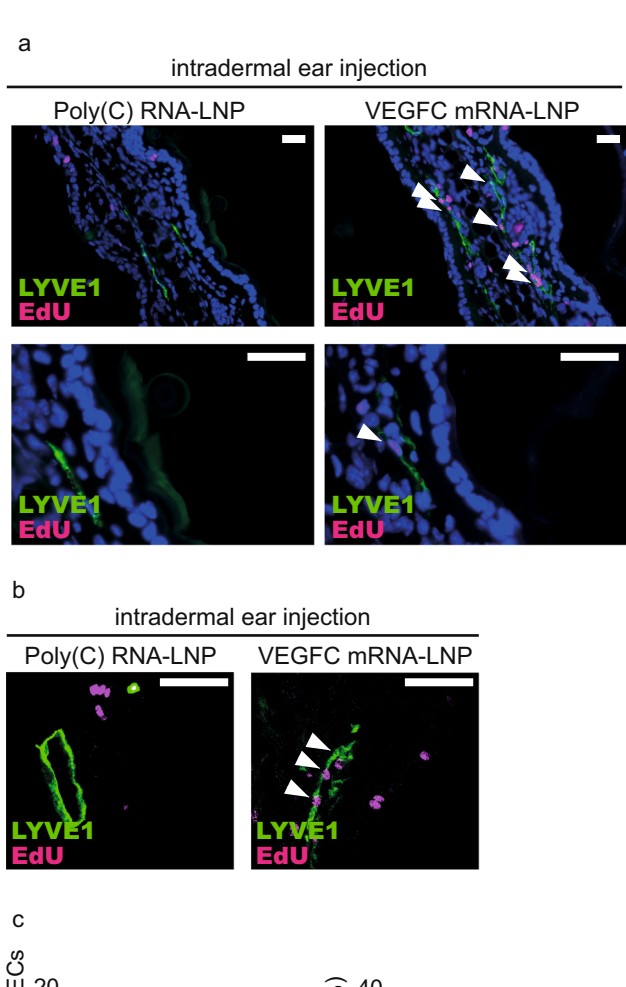

**Fig. 3 Administration of VEGFC mRNA-LNPs induces local lymphatic proliferation in mice in vivo. a–b** Assessment of lymphatic proliferation 5 days after intradermal injection with 1 μg of Poly(C) or VEGFC mRNA-LNPs. EdU staining and anti-LYVE1 immunostaining of slides processed by paraffin-based histology (bars, 50 μm) shown by widefield (**a**) and confocal imaging (**b**). Arrows indicate EdU and LYVE1 double positive lymphatic endothelial cell (LEC) nuclei. Representative images of 3 ears of 3 mice in each group are shown. **c** Number of EdU positive LECs (two-tailed, unpaired *T*-test, *P* = 0.0260 for 3 mice) and mitotic index (two-tailed, unpaired *T*-test, *P* = 0.0266 for 3 mice) are shown 5 days after intradermal injection with 1 μg of Poly(C) or VEGFC mRNA-LNPs. Quantitative data for lymphatic proliferation are represented as mean and SEM from 3 injected ears of 3 mice in each group.

**VEGFC mRNA-LNP administration reverses experimental lymphedema in mice**. After setting up and validating the previously described experimental system[46], Poly(C) or VEGFC mRNA-LNPs were intradermally injected into the hind paw of tamoxifen and Diphtheria toxin-treated Flt4-CreER^T2; *iDTR^fl/fl* mice 8 days after the first treatment with Diphtheria toxin. Of note, the Poly(C) and VEGFC mRNA-LNPs were injected into the contralateral paw of the same animal. These experiments

revealed that the administration of VEGFC mRNA-LNPs effectively reduced paw thickness and clinical score in the experimental secondary lymphedema mouse model compared to the Poly(C) RNA-LNP control treatment (Fig. 9a). Importantly, VEGFC mRNA-LNP injection effectively reduced the fibroadipose area in Haematoxylin and Eosin stained sections and Collagen I content compared to the Poly(C) RNA-LNP control 30 and 75 days after the treatment (Fig. 9b–c, f). VEGFC mRNA-LNP also induced the growth of the lymphatic vessels (Fig. 9d–e). This newly formed lymphatic network appeared to be fully functional because drainage of labeled macromolecules to the popliteal lymph node was enhanced at 30 and 75 days after the treatment compared to the Poly(C) RNA-LNP control injection (Fig. 10a–b). Taken together, our results indicate that nucleoside-modified VEGFC mRNA-LNP treatment reverses experimental lymphedema in a validated mouse model without showing any adverse events in the experimental animals.

## Discussion

Nucleoside-modified mRNA-LNP is a novel, highly effective, and safe therapeutic modality that is used for vaccine development, protein replacement therapy, and gene editing in preclinical models and human trials[34–38,40,41]. Importantly, nucleoside-modified SARS-CoV-2 mRNA-LNP vaccines developed by Moderna and Pfizer/BioNTech have been approved for mass vaccination in multiple countries. In this study we demonstrated that nucleoside-modified and Fast Protein Liquid Chromatography (FPLC)-purified murine VEGFC-encoding mRNA-LNPs were highly efficient to stimulate VEGFC protein secretion in vitro and in vivo, and most importantly, induced durable growth of lymphatic vessels in various organs in mice (Figs. 1–3 and Supplementary Figs. 2, 3, and 5). This is a critical finding because in addition to some well-known limitations of protein therapeutics (high cost of production, requirement for complicated cell culture, and protein purification systems), previous reports showed that VEGFC protein treatment had only limited effect on lymphatic growth in various organs, most likely due to the insufficient duration of activity[5,19,20,22]. In connection, our comparative studies revealed that nucleoside-modified VEGFC mRNA-LNPs were much more efficient to trigger lymphatic proliferation than recombinant VEGFC protein treatment (Figs. 2–3, Supplementary Figs. 3–5).

Safety is a critical requirement for any therapeutic agent. The use of nucleoside-modified mRNA has several important advantages over other types of therapeutic protein delivery systems, such as the lack of integration into the host genome, no anti-vector immunity, transient and thus highly controllable protein production, and the lack of activation of strong inflammatory responses. First, the organ-specific effect of the mRNA-LNP platform was demonstrated in vivo. Strong local GFP expression was detected at the injection site after intradermal GFP mRNA-LNP treatment, but not in other organs including the lung and the small intestine in the same animal (Supplementary Fig. 6). These first experiments confirmed local protein production and ruled out potential off-target effects after intradermal VEGFC mRNA-LNP delivery. These results are in agreement with our previous report[39], in which a luciferase reporter assay was used to compare protein production from mRNA-LNPs after administration via various routes. Similarly, VEGFC-induced lymphangiogenesis was limited to the injection sites in mice intradermally treated with VEGFC mRNA-LNPs and it was not detectable in other anatomical sites (Fig. 4).

To demonstrate another aspect of safety, we found that VEGFC mRNA-LNP treatment resulted in non-significant increase in

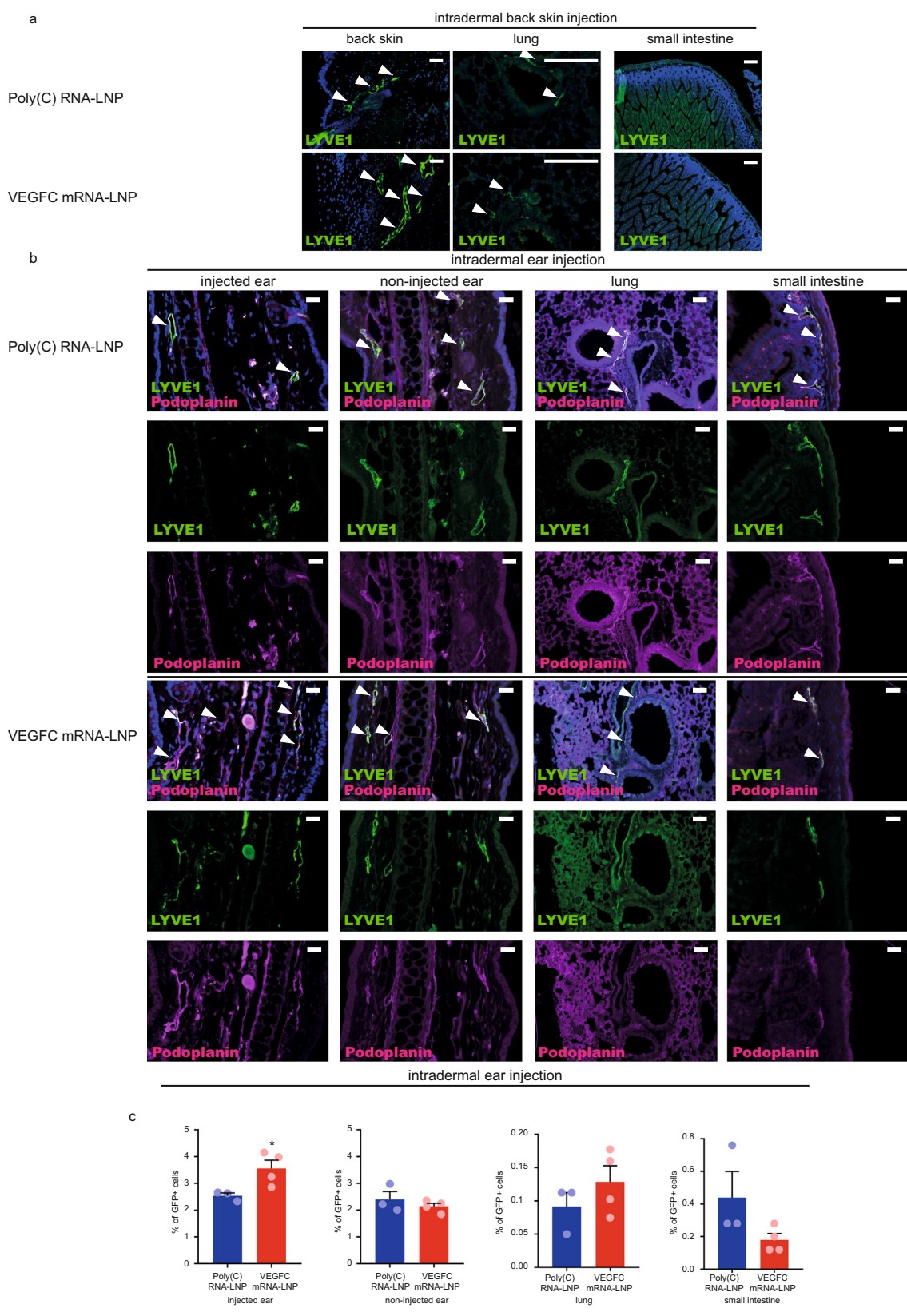

blood vessel formation and immune cell infiltration compared to Poly(C) RNA-LNP delivery (Fig. 5, Supplementary Fig. 7). Our findings are in accordance with previous studies which demonstrated that VEGFC, in contrast to VEGFD which induces both blood and lymph vessel growth in adults, did not have a major impact on blood vessel proliferation (Fig. 5a–b, Supplementary Fig. 7)[28,48–50]. VEGFC was reported to induce some proliferation and dilation of blood vessels and a slight increase in their permeability after viral vector delivery in a part of the studies[24,25,28,50,51]. It is possible that some of these effects were

**Fig. 4 Administration of VEGFC mRNA-LNPs results in organ-specific effects in vivo. a** 1 µg of Poly(C) or VEGFC mRNA-LNPs were intradermally injected into the back skin of C57Bl/6 mice. Growth of lymphatic vessels in back skin, lung, and small intestine was monitored 22 days after the injection. Representative images 22 days after the treatment of the back skin of 4 mice per group are shown by anti-LYVE1 immunostaining of slides processed by paraffin-based histology (bars, 125 µm). Arrows indicate LYVE1 positive lymphatic vessels. **b** 5 µg of Poly(C) or VEGFC mRNA-LNPs were intradermally injected into the ear of C57Bl/6 mice. Growth of lymphatic vessels in the injected and non-injected ears, lung, and small intestine was assessed 22 days after the injection. Representative images after the ear treatment of 4 mice per group are shown by anti-LYVE1 and anti-Podoplanin immunostaining of slides processed by paraffin-based histology (bars 125 µm). Arrows indicate LYVE1 and Podoplanin double positive lymphatic vessels. **c** Monitoring the effect of intradermal injection into ears of *Prox1*[GFP] mice with 5 µg of Poly(C) or VEGFC mRNA-LNPs shown by flow cytometry analysis. 5 µg of Poly(C) or VEGFC mRNA-LNPs were intradermally injected into the ear. Injected and non-injected ears, lungs, and small intestines were harvested and digested into single cell suspension. Quantitative data for GFP positive cell number are represented as mean and SEM from Poly(C) or VEGFC mRNA-LNP-injected ears, non-injected ears, non-injected lungs, and non-injected small intestines of 4 mice in VEGFC mRNA-LNP group and 3 mice in control group 22 days after mRNA-LNP injection (two-tailed, unpaired *T*-test for injected ears of Poly(C) RNA-LNP injected vs VEGFC mRNA-LNP injected mice $P = 0.0402$, two-tailed, unpaired *T*-test for contralateral non-injected ears of Poly(C) RNA-LNP injected vs. VEGFC mRNA-LNP injected mice $P = 0.4006$, two-tailed, unpaired *T*-test for lungs of Poly(C) RNA-LNP injected vs VEGFC mRNA-LNP injected mice $P = 0.3168$, two-tailed, unpaired *T*-test for small intestines of Poly(C) RNA-LNP injected vs VEGFC mRNA-LNP injected mice $P = 0.1252$). Asterisks indicate $P < 0.05$ compared with control. All cell nuclei are labeled with DAPI (blue) in paraffin-embedded tissues.

not induced by the VEGFC directly, but they were elicited by the adenoviral delivery of the lymphangiogenic factor, or require special conditions. We also detected a transient and slight increase in blood vessel permeability shortly after VEGFC mRNA-LNP treatment compared to Poly(C) RNA-LNP delivery suggesting a possible transient effect of VEGFC on blood vessel permeability but this finding needs to be further investigated in the future. It should be noted that nucleoside-modified VEGFC mRNA-LNP treatment may induce the stimulation of VEGFR2, but this process does not lead to significant proliferation of blood vessels. Further studies will be needed to understand the effect of our platform on VEGFR2 signaling. Prior studies demonstrated that the VEGFC protein might act as a chemoattractant for monocytes and macrophages[52,53], but the nucleoside-modified VEGFC mRNA-LNP platform did not increase immune cell infiltration significantly compared to Poly(C) RNA-LNP administration 22 days after the treatment (Fig. 5c–d). Collectively, our results indicate that our novel platform lacks the development of serious adverse effects while driving organ-specific lymphatic growth. It is important to note that our safety studies have focused on VEGFC-induced adverse events and have not investigated the potential effect of LNP delivery on blood vessel formation and immune cell infiltration that will be addressed in future studies.

Our results indicate that the newly formed lymphatic vessels, induced by nucleoside-modified VEGFC mRNA-LNP administration into the ear and hind limb, show normal morphology and are ready to take up and transport macromolecules to the local lymph nodes (e.g., popliteal) without any sign of leakage even after an extended period of time (up to 75 days) (Fig. 6a, b). This is an important finding because some studies using AAV VEGFC constructs stimulated the growth of lymphatic endothelial marker positive cells, which did not form vessel-like structures, especially in the gastrocnemius muscle[31,54]. In addition, an increase in the number of lymphatic vessels with smooth muscle coverage in the VEGFC mRNA-LNP treated animals was detected compared to the Poly(C) RNA-LNP control (Fig. 6c–e) raising the possibility that VEGFC mRNA-LNP treatment may also induce the proliferation of collecting lymphatics, but further studies will be needed to examine this interesting finding in detail.

To evaluate the effect of nucleoside-modified VEGFC mRNA-LNP treatment in an experimental disease model, a recently developed[46] Diphtheria toxin-inducible genetic secondary lymphedema mouse model was set up and optimized. In the rigorously validated experiments secondary lymphedema development was documented by paw thickness, clinical score, fibroadipose area, Collagen I, and lymphatic function

measurements (Figs. 7–8). In accordance with prior studies we also only detected local lymphatic deletion after intradermal injection of Diphtheria Toxin with no obvious systemic effects[46]. Importantly, a single low-dose (1 µg) of VEGFC mRNA-LNPs effectively reduced paw thickness, clinical score, the fibroadipose area, and Collagen I content compared to the Poly(C) RNA-LNP control 30 and 75 days after disease induction in the experimental secondary lymphedema mouse model after injecting into the contralateral paw of the same animal (Fig. 9a–c, f). VEGFC mRNA-LNP stimulated the formation of a functional network of lymphatic vessels (Fig. 9d–e), which was efficient to drain macromolecules from the paw to the popliteal lymph nodes in the VEGFC mRNA-LNP treated animals 30 and 75 days after Diphtheria toxin injection (Fig. 10a–b) without showing adverse events in the experimental animals. These important results demonstrate that administration of a single low-dose of VEGFC mRNA-LNPs is an efficient strategy to induce the formation of functional lymphatic vessels, which then reverse the development and progression of secondary lymphedema (Figs. 9–10). Further studies (e.g., in large animal models) will be needed to develop the most efficient treatment strategies and protocols to cure secondary lymphedema utilizing the nucleoside-modified VEGFC mRNA-LNP platform.

An adenovirus-based VEGFC delivery platform (Lymfactin) will be one potential treatment option (entered Phase II clinical trial (NCT03658967)) for secondary lymphedema[25,55]. However, adenovirus or AAV-based delivery systems have well-known drawbacks including the potential chance for genomic integration, anti-vector immune responses, difficulties with regulation and toxicity as discussed above. We demonstrated that the novel nucleoside-modified VEGFC mRNA-LNP platform is an effective tool for the delivery of the lymphangiogenic factor, and importantly, it has a number of important beneficial features. First, VEGFC mRNA-LNPs cannot integrate into the genome and therefore, there is no risk of genomic instability and cancer formation, which makes it a safe therapeutic modality to use in humans. Second, the VEGFC mRNA-LNP platform induces the formation of a fully functional lymphatic network. Third, the expression level of the lymphangiogenic factor is scalable and controllable. A novel gain of function approach for identifying the organ-specific physiological and pathophysiological roles of the lymphatic system was designed and evaluated. Fourth, and most importantly, the nucleoside-modified VEGFC mRNA-LNP platform proved to be effective in reversing disease progression in an experimental secondary lymphedema mouse model by inducing the formation of a functional lymphatic network.

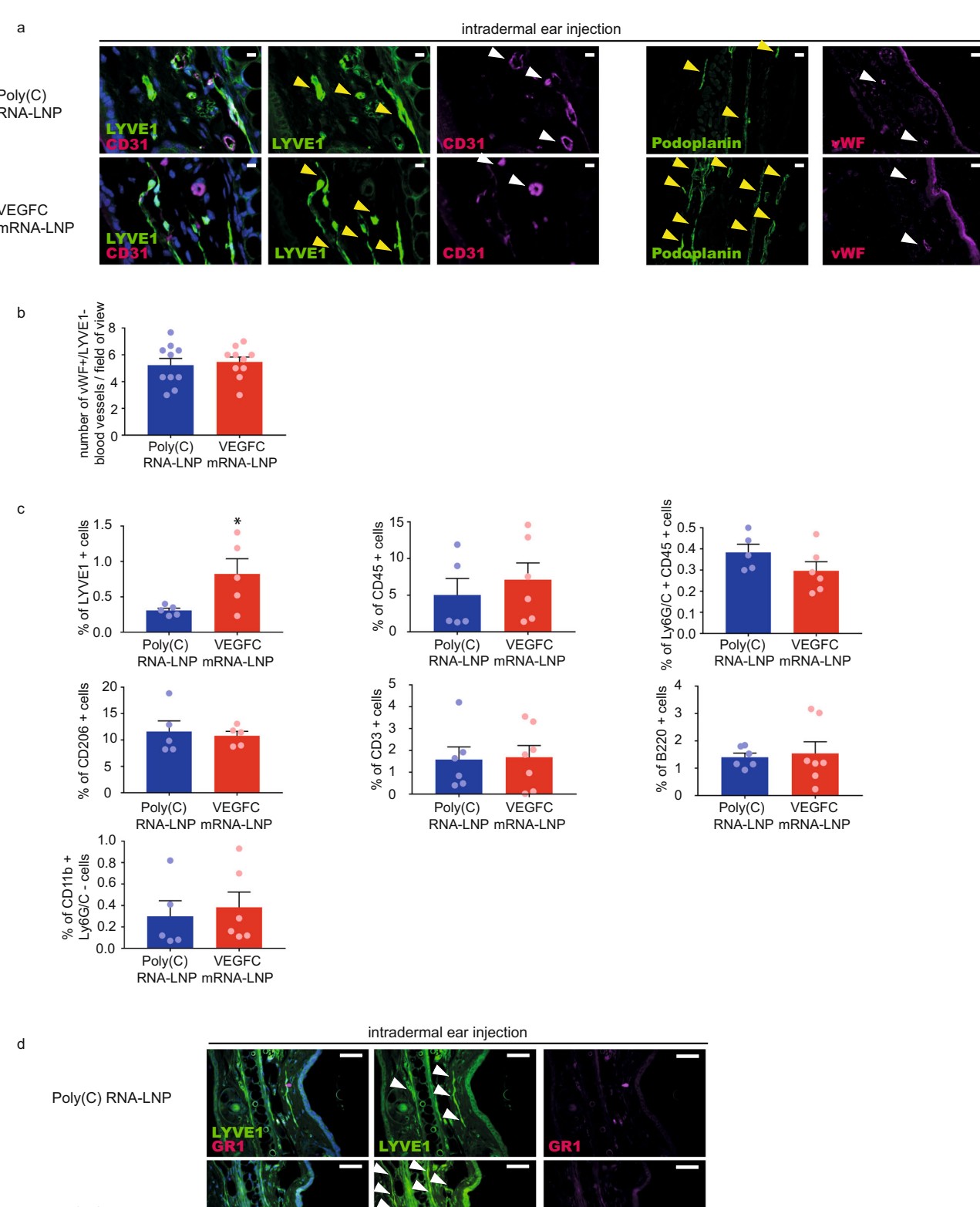

Collectively, our study describes the use of the nucleoside-modified mRNA in the field of lymphatics and based on the data presented here we believe that the application of this platform is a viable strategy for the treatment of lymphedema. Our results represent a major advancement in the fields of mRNA therapy, vascular biology, and lymphatics.

## Methods

**Experimental animals**. 6–12-week-old C57Bl/6 mice were purchased from commercial sources, *Prox1GFP* BAC lymphatic reporter animals obtained from the Mutant Mouse Regional Resource Centers (MMRRC) were maintained in heterozygous form and genotyped by a transgene-specific PCR using 5′ - GAT GTG CCA TAA ATC CCA GAG CCT AT – 3′forward and 5′ – GGT CGG GGT AGC GGC TGA A – 3′ reverse primers[42]. To eliminate the lymphatic endothelial cells in

**Fig. 5 Administration of VEGFC mRNA-LNPs results in non-significant blood vessel proliferation and immune response activation compared to the Poly(C) RNA-LNP control. a** 1 µg of Poly(C) or VEGFC mRNA-LNPs was intradermally injected into the ear of *C57Bl/6* mice and the growth of blood and lymphatic vessels were examined 22 days after the injection. Representative images of anti-CD31, anti-LYVE1, anti-Podoplanin, and anti-vWF stained paraffin-embedded ear samples are shown of 5 mice in each group (bars, 25 µm (anti-CD31, anti-LYVE1), 50 µm (anti-Podoplanin, anti-vWF)). Yellow arrows indicate LYVE1 positive lymphatic vessels, white arrows indicate CD31 positive and LYVE1 negative blood vessels. **b** The number of vWF high and LYVE1 negative blood vessels was determined 22 days after the administration of 1 µg of Poly(C) or VEGFC mRNA-LNPs. Data are represented as mean and SEM from slides of ears of 10 mice per group (two-tailed, paired *T*-test, *P* = 0.6344 for 10 mice). **c** Monitoring the effect on lymphatic endothelial cells and immune cells after intradermal injection of 1 µg of Poly(C) or VEGFC mRNA-LNPs into ears of *Prox1*[GFP] mice shown by flow cytometry analysis. Quantitative data for LYVE1+, CD45+, Ly6G/C (GR1)+, CD206+, CD3+, B220+ and CD11b+ Ly6G/C- cell numbers are represented as mean and SEM from Poly(C) or VEGFC mRNA-LNP-injected ears of 5–7 mice in each group, 22 days after mRNA-LNP injection (two-tailed, unpaired *T*-test, *P* = 0.0450 for LYVE1+ for 5 mice injected with Poly(C) and for 5 mice injected VEGFC mRNA-LNP, *P* = 0.5365 for CD45+ for 5 mice injected with Poly(C) and for 6 mice injected VEGFC mRNA-LNP, *P* = 0.1689 for Ly6G/C+ and CD45+ for 5 mice injected with Poly(C) and for 6 mice injected VEGFC mRNA-LNP, *P* = 0.7211 for CD206+ for 5 mice injected with Poly(C) and for 5 mice injected VEGFC mRNA-LNP, *P* = 0.8947 for CD3+ for 6 mice injected with Poly(C) and for 7 mice injected VEGFC mRNA-LNP, *P* = 0.7748 for B220+ for 6 mice injected with Poly(C) and for 7 mice injected VEGFC mRNA-LNP and *P* = 0.6922 for CD11b+ Ly6G/C- for 5 mice injected with Poly(C) and for 6 mice injected VEGFC mRNA-LNP comparing cell number). **d** Ly6G/C (GR1) positive immune cells were visualized after intradermal administration of 1 µg of Poly(C) or VEGFC mRNA-LNPs into the ears shown by anti-GR1 and anti-LYVE1 22 days after the injection. Representative images of anti-GR1 stained paraffin-embedded ear samples are shown of 3 mice per group (bars, 50 µm). Arrows indicate LYVE1 positive lymphatic vessels. Asterisks indicate *P* < 0.05 compared with control. All cell nuclei are labeled with DAPI (blue) in paraffin-embedded tissues.

a genetic experimental lymphedema model the Flt4-CreER[T2]; *iDTR*[fl/fl] strain on the C57Bl/6 background was used[45,46]. Allele-specific PCR was performed for the genotyping the Flt4-CreER[T2]; *iDTR*[fl/fl] strain using Flt4-CreER[T2] knock in-specific forward 5′ – GGCTGGACCAATGTAAATATTG – 3′) and reverse (5′ – CATCA TCGAAGCTTCACTG – 3′), Flt4 wild type-specific forward (5′ – CACTATGC TCCGTGTCTTG – 3′) and reverse (5′ – GTGACTCTCAGACATATG – 3′) and *iDTR*[fl/fl] allele-specific reverse (5′ – CAT CAA GGA AAC CCT GGA CTA CTG – 3′), common forward (5′ – AAA GTC GCT CTG AGT TGT TAT – 3′) and wild type site-specific reverse (5′ – GGA GCG GGA GAA ATG GAT ATG – 3′) primers. The sequences of all genotyping primer sets are shown in Supplementary Table 1. All animal experiments were approved by the Animal Experimentation Review Board of the Semmelweis University and the Government Office for Pest County (Hungary). Experimental animals were housed in either specific pathogen free or conventional animal facilities between 18–22 °C, 45% humidity, and 12/12 hours dark–light cycles.

**Design and production of GFP and VEGFC mRNAs.** mRNAs were produced as previously described using T7 RNA polymerase (Megascript, Ambion) on linearized plasmids encoding codon-optimized mouse Vascular Endothelial Growth Factor C (pTEV-muVEGFC-A101) and green fluorescent protein (pTEV-GFP-A101). mRNAs were transcribed to contain 101 nucleotide-long poly(A) tails[56]. One-methylpseudouridine (m1Ψ)-5′-triphosphate (TriLink) instead of UTP was used to generate modified nucleoside-containing mRNA. RNAs were capped using the m7G capping kit with 2′-O-methyltransferase (ScriptCap, CellScript) to obtain cap1. mRNA was purified by Fast Protein Liquid Chromatography (FPLC) (Akta Purifier, GE Healthcare), as described[57]. All mRNAs were analyzed by agarose gel electrophoresis and were stored frozen at −20 °C.

**LNP formulation of Poly(C), GFP, and VEGFC mRNAs.** Poly(C) (Sigma-Aldrich), m1Ψ-containing GFP, and murine VEGFC-encoding mRNAs were encapsulated in LNPs using a self-assembly process in which an aqueous solution of mRNA at pH = 4.0 is rapidly mixed with a solution of lipids dissolved in ethanol[58]. LNPs used in this study were similar in composition to those described previously[58,59], which contain an ionizable cationic lipid (proprietary to Acuitas)/phosphatidylcholine/cholesterol/PEG-lipid (50:10:38.5:1.5 mol/mol) and were encapsulated at an RNA to total lipid ratio of ~0.05 (wt/wt). They had a diameter of ~80 nm as measured by dynamic light scattering using a Zetasizer Nano ZS (Malvern Instruments Ltd, Malvern, UK) instrument. mRNA-LNP formulations were stored at −80 °C at a concentration of mRNA of 1 µg/µl. All mRNA-LNP formulations for this study were provided after rigorous quality control (size and integrity) to avoid batch-to-batch variations.

**In vitro experiments.** HEK293T cells (300,000 cells) were cultured for 24 h and then transfected with 1 µg of Poly(C), GFP, or VEGFC mRNA-LNPs by adding them to the serum-free culture medium. GFP expression of GFP mRNA-LNP transfected cells were visualized by a DMI 6000B Leica fluorescent inverted microscope (acquisition was performed using LAS X Software (Leica) 7.4.2.18368). At different time points after 1 µg of Poly(C), GFP or VEGFC mRNA-LNP treatment of HEK293T cells, the supernatant of the samples were harvested and the cells were lysed in RIPA Lysis Buffer (Thermo Fisher Scientific) supplemented with protease and phosphatase inhibitors, as described previously[60].

**Monitoring in vivo lymphatic growth in experimental mice.** 6–12-week-old wild type or *Prox1*[GFP] lymphatic reporter mice were injected with 0.04, 0.2, 1, or 5 µg of Poly(C), GFP or VEGFC mRNA-LNPs by intradermal ear, back skin and paw, intratracheal, intraperitoneal, and intramuscular (m. gastrocnemius) injections. In most of the experiments, the contralateral ear, back skin, paw, and m. gastrocnemius injection was used as control. In parallel experiments, 1 µg of human recombinant VEGFC protein (R&D Systems, 2179-VC) was administrated intradermally into the ear and back skin of 6–12-week-old *Prox1*[GFP] lymphatic reporter animals. Of note, it is well documented that adenovirus-based delivery of the human VEGFC induces lymphatic proliferation in mouse models in vivo indicating the effectiveness of the human protein in mice[23,27]. After various incubation times (2–60 days) the animals were sacrificed, and the tissues were harvested. Lymphatic growth in lymphatic reporter animals was visualized by fluorescent stereo microscopy using a Nikon SMZ25 microscope connected to a Nikon DS-Ri2 camera. Total length of the lymphatic network, diameter, and branching points per vision field were quantified based on stereo microscopic pictures as described below. Ear, back skin, paw, lung, and small intestine of Poly(C) or VEGFC mRNA-LNP injected animals were harvested, fixed, and processed for paraffin-based histology.

**Monitoring experimental lymphedema in vivo.** A genetic secondary lymphedema model was used which was developed previously[46]. Tamoxifen (Sigma-Aldrich) was dissolved at 20 mg/ml in corn oil (Sigma-Aldrich), and 112.5 mg/kg tamoxifen was injected into 8–24-week-old Flt4-CreER[T2]; *iDTR*[fl/fl] mice[46] daily for 5 days intraperitoneally. One week after the last tamoxifen injection the animals were injected with 60 ng of Diphtheria toxin (Sigma-Aldrich) daily for 3 days into the hind paw. As a control, PBS (Lonza) was administered into the contralateral paw of the same mouse. In the second set of experiments eight days after the first Diphtheria toxin treatment of both paws, animals were injected with 1 µg of Poly (C) RNA-LNP into one paw and 1 µg of VEGFC mRNA-LNP into the contralateral paw of the same mouse. To monitor lymphedema development, the thickness of the paw was assessed by spring loaded custom caliper (Kroeplin). Visible clinical signs of lymphedema were scored on a 0–10 scale by two investigators blinded for the origin and treatment of the mice. The paws were scored based on their swelling (0–4), thickness (0–2), color (0–2), and skin tightness (0–2). Tissues were harvested for histology after 30 and 75 days of Diphtheria toxin injection, and immunostaining using lymphatic and collagen markers and Haematoxylin and Eosin staining were performed. 30 and 75 days after the first Diphtheria toxin injection the function of the lymphatic vessels was tested using intradermally injected 70 kDa Rhodamine Dextran (Life Technologies) as described below.

**Monitoring of VEGFC expression using Western blot and ELISA analyses.** The Triton-soluble fraction of the lysates and the supernatants were boiled in sample buffer, run on SDS-PAGE, and immunoblotted with antibodies in a dilution of 1:1,000 against GFP (Abcam, ab6673), VEGFC (R&D Systems, AF752), and β-actin (Sigma-Aldrich, A2228). HRP-labeled, anti-mouse, and anti-goat secondary antibodies were used in a dilution of 1:10,000 followed by the application of PICO (Thermo Scientific) or ECL (Amersham, GE Healthcare) with detection by an Azure Biosystems c600 imaging system.

VEGFC protein expression in cell lysates and supernatants of the in vitro experiments was determined at different time points including 8 hours, 1, 2, 4, 8, and 12 days using a VEGFC Rat ELISA Kit (Invitrogen, BMS626-2) following the instructions of the manufacturer. To determine VEGFC protein expression in vivo at 1, 5, 10, 15, and 20 days after intradermal Poly(C) mRNA-LNP or VEGFC mRNA-LNP injection ear samples were digested with the Liberase II kit (Roche)

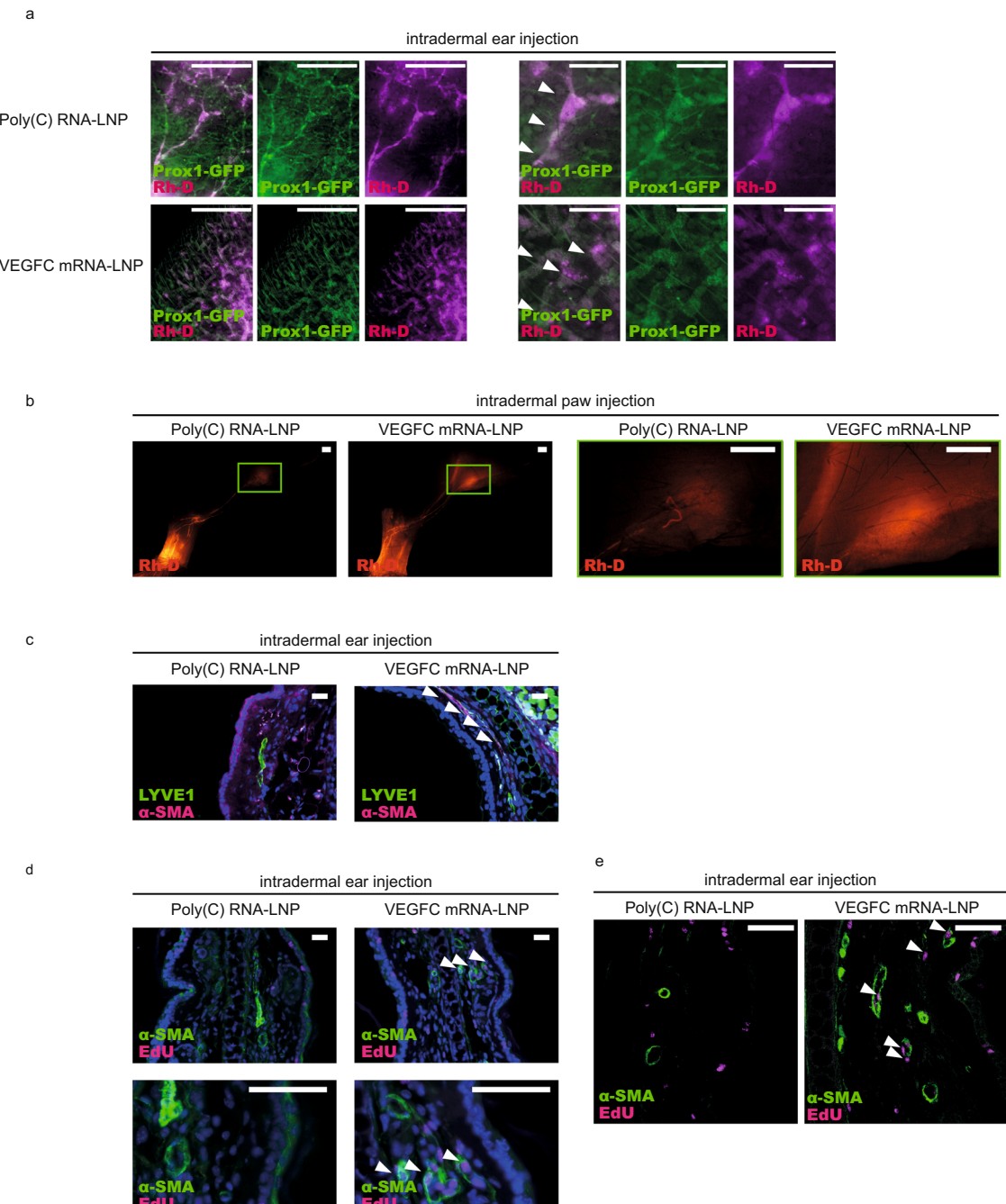

**Fig. 6 Administration of VEGFC mRNA-LNPs stimulates active lymphatic function. a** Monitoring active lymphatic function in the ears of *Prox1^GFP* mice after intradermal injection of 1 μg of Poly(C) or VEGFC mRNA-LNPs. Twenty-two days after treatment, 70 kDa Rhodamine dextran (Rh-D) was injected into the ear and the transport of the molecule was monitored by fluorescent microscopy 60 min post Rh-D administration. Representative images are shown of 10 injected mouse ears per group (bars, 1000 μm). Arrows indicate Prox1-GFP positive and Rh-D positive lymphatic vessels. **b** Monitoring active lymphatic function in the paws of mice after intradermal injection of 1 μg of Poly(C) or VEGFC mRNA-LNPs. Seventy-five days after treatment, 70 kDa Rh-D was injected into the paws and the transport of the molecule was monitored by fluorescent microscopy 90 min post Rh-D administration. Representative images are shown of 2 mouse hind limbs in each group (bars, 1000 μm). Green rectangles show the magnified area which represent the area of popliteal lymph nodes. **c** Analysis of lymphatic morphology in the ear of mice injected with 1 μg of Poly(C) or VEGFC mRNA-LNPs. Representative images 22 days after the treatment of 3 mouse ears per group are shown by anti-LYVE1 and anti-α-SMA immunostaining of slides processed by paraffin-based histology (bars, 50 μm). Arrows indicate LYVE1 low lymphatics surrounded by α-SMA positive cells. **d–e** Assessment of lymphatic proliferation 5 days after intradermal injection with 1 μg of Poly(C) or VEGFC mRNA-LNPs. Representative images of 3 mouse ears are shown in each group with EdU staining and anti-α-SMA immunostaining of slides processed by paraffin-based histology (bars, 50 μm) and shown by widefield (**d**) and confocal imaging (**e**). Arrows indicate EdU positive cells surrounded by α-SMA positive cells. All cell nuclei are labeled with DAPI (blue) in paraffin-embedded tissues.

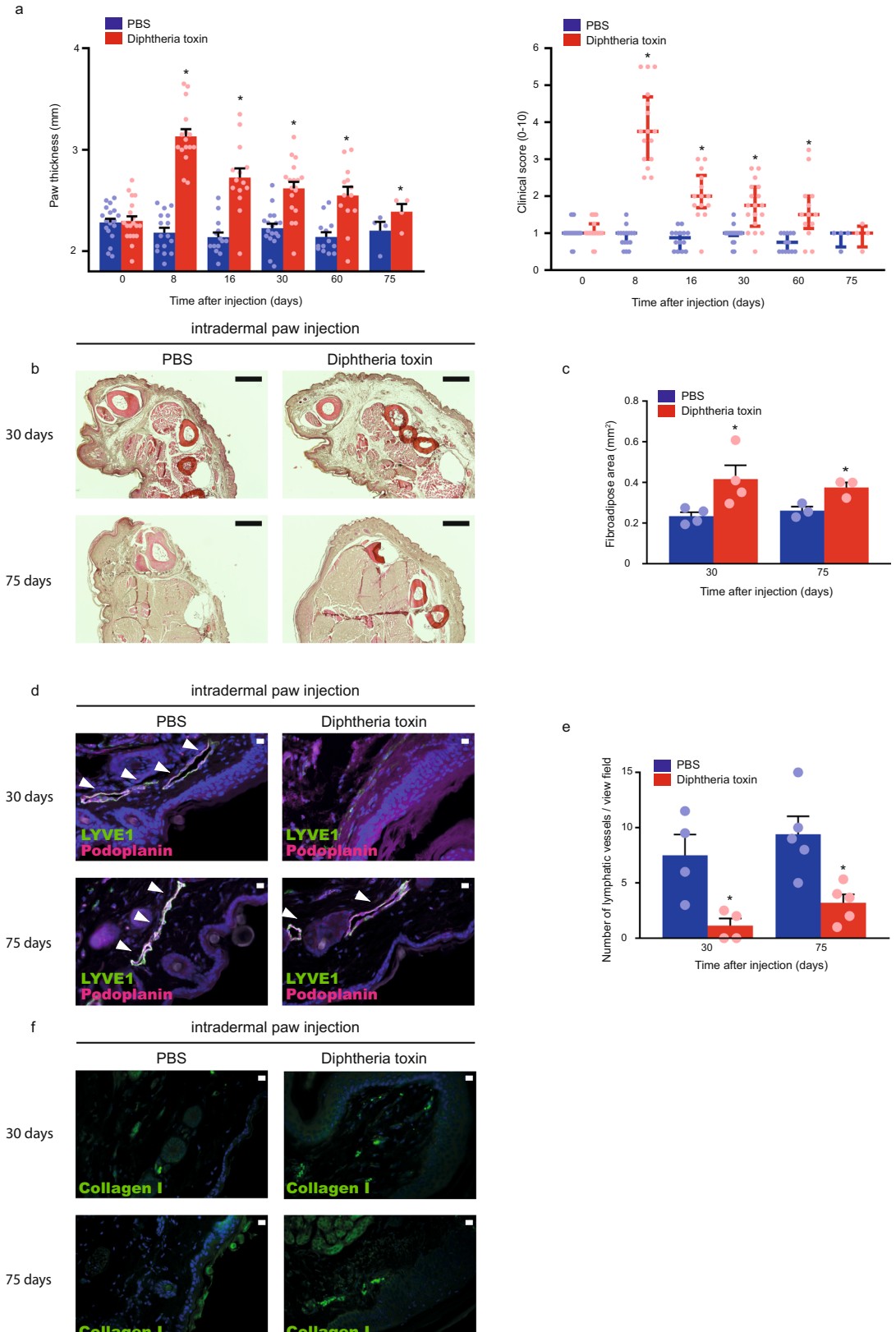

and the secreted amount of VEGFC in the interstitial fluid of ear was determined using the same ELISA Kit described above. All investigators performing Western Blot or ELISA assays were blinded for the sample origin.

**Whole-mount immunostaining**. The 2% paraformaldehyde (Sigma-Aldrich) fixed ears of *Prox1*^GFP lymphatic reporter mice injected with Poly(C) RNA-LNP or VEGFC

mRNA-LNPs were imaged for GFP fluorescence and anti-von Willebrand Factor (anti-vWF) (Sigma-Aldrich, F3520, 1:100) labeling after overnight incubation, which was followed by anti-rabbit secondary antibody staining conjugated to Alexa Fluor 568 (Life Technologies, AF11036, 1:250). Thereafter confocal microscopy was performed using a Nikon Eclipse A1 Confocal Laser Microscope and a Nikon SMZ25 stereo microscope connected to a Nikon DS-Ri2 camera (Nikon Instruments).

**Fig. 7 Diphtheria toxin induces experimental secondary lymphedema development in Flt4-CreER$^{T2}$; iDTR$^{fl/fl}$ mice. a** Monitoring paw thickness and paw clinical score in tamoxifen-treated Flt4-CreER$^{T2}$; iDTR$^{fl/fl}$ mice after treatment with PBS or Diphtheria toxin. Quantitative data are represented as mean and SEM for paw thickness and median and IQR for clinical score from 4–18 mouse paws in each group (two-tailed, paired $T$-test for paw thickness $P = 4.21 \times 10^{-12}$ on day 8 for 16 mice, $P = 5.05 \times 10^{-6}$ on day 16 for 14 mice, $P = 1.62 \times 10^{-5}$ on day 30 for 18 mice, $P = 0.0008$ on day 60 for 13 mice, and $P = 0.0112$ on day 75 for 4 mice. Two-tailed Wilcoxon signed-rank test for clinical score $P = 3.05 \times 10^{-5}$ on day 8 for 16 mice, $P = 0.0002$ on day 16 for 14 mice, $P = 6.10 \times 10^{-5}$ on day 30 for 18 mice, $P = 0.0020$ on day 60 for 13 mice, and $P > 0.9999$ on day 75 for 4 mice). **b** Haematoxylin and Eosin staining of the paws of Flt4-CreER$^{T2}$; iDTR$^{fl/fl}$ tamoxifen-treated mice 30 and 75 days after treatment with PBS or Diphtheria toxin. Representative images are shown of 5 mouse paws per group (bars, 200 μm). **c** Paw fibroadipose area of tamoxifen-treated Flt4-CreER$^{T2}$; iDTR$^{fl/fl}$ mice 30 and 75 days after intradermal paw injection with PBS or Diphtheria toxin. Quantitative data are represented as mean and SEM from Haematoxylin and Eosin stained slides of 3–4 mouse paws in each group (two-tailed, paired $T$-test, $P = 0.0469$ on day 30 for 4 mice and $P = 0.0172$ on day 75 for 3 mice). **d** Representative images of anti-LYVE1 and anti-Podoplanin immunostaining of the paws of Flt4-CreER$^{T2}$; iDTR$^{fl/fl}$ tamoxifen-treated mice 30 and 75 days after treatment with PBS or Diphtheria toxin. Representative images are shown of 4–5 mouse paws in each group (bars, 50 μm). Arrows indicate LYVE1 and Podoplanin double positive lymphatic vessels. **e** Number of lymphatic vessels of tamoxifen-treated Flt4-CreER$^{T2}$; iDTR$^{fl/fl}$ mice 30 and 75 days after intradermal paw injection with PBS or Diphtheria toxin. Quantitative data are represented as mean and SEM from 4–5 mouse paws in each group (two-tailed, paired $T$-test, $P = 0.0319$ after 30 days for 4 mice and $P = 0.0434$ after 75 days for 5 mice). **f** Representative images of anti-Collagen I immunostaining of the paws of Flt4-CreER$^{T2}$; iDTR$^{fl/fl}$ tamoxifen-treated mice 30 and 75 days after treatment with PBS or Diphtheria toxin. Representative images are shown of 5 mouse paws in each group (bars, 50 μm). Asterisks indicate $P < 0.05$ compared with control. All cell nuclei are labeled with DAPI (blue) in paraffin-embedded tissues.

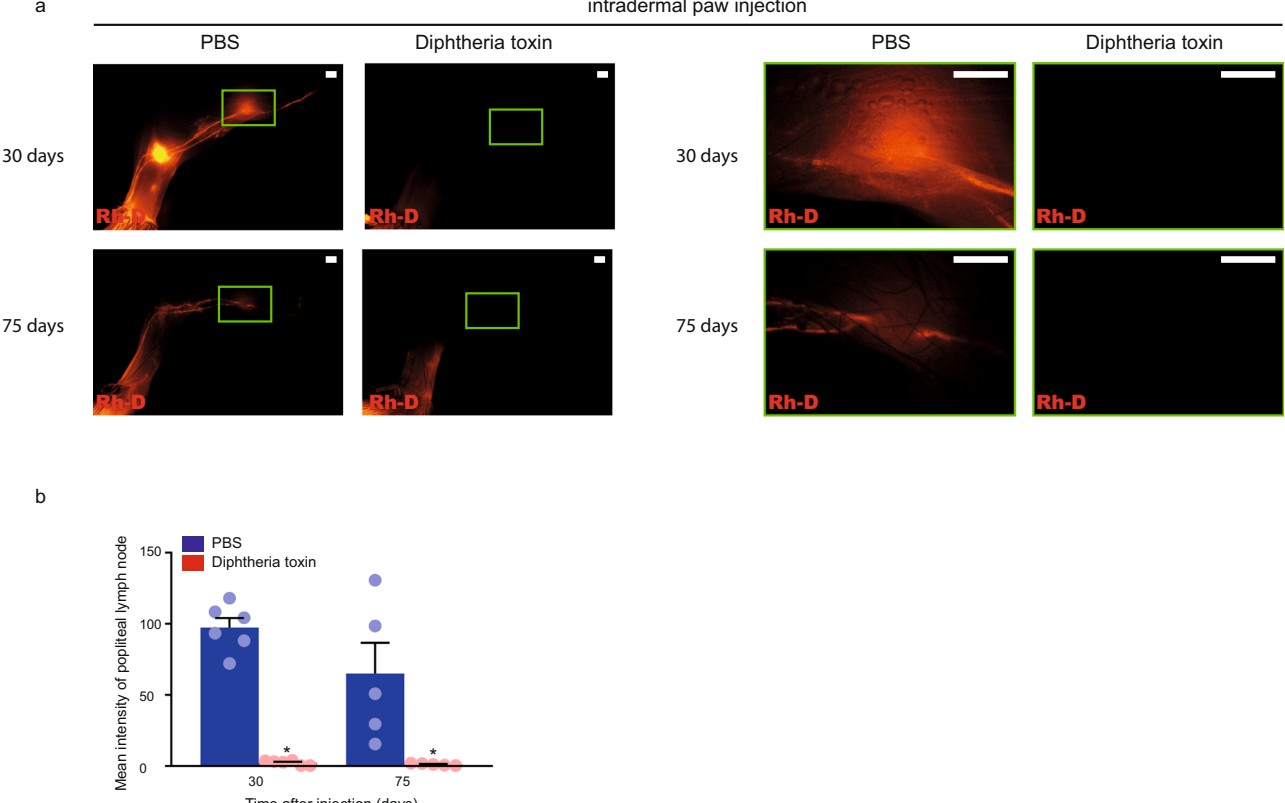

**Fig. 8 Administration of Diphtheria toxin into the paws of tamoxifen-treated Flt4-CreER$^{T2}$; iDTR$^{fl/fl}$ animals reduces lymphatic function as the result of secondary lymphedema development. a** Monitoring active lymphatic function in the contralateral hind limbs of the same tamoxifen-treated Flt4-CreER$^{T2}$; iDTR$^{fl/fl}$ mouse 30 or 75 days after intradermal injection of PBS into one paw and Diphtheria toxin into another paw. 70 kDa Rh-D was injected into the paws and the transport of the molecule was monitored by fluorescent microscopy 90 min post Rh-D administration. Representative images are shown of 5–6 mouse hind limbs in each group. (bars, 1000 μm). Green rectangles show the magnified area which represent the area of popliteal lymph nodes. **b** Fluorescent intensity of popliteal lymph node of tamoxifen-treated Flt4-CreER$^{T2}$; iDTR$^{fl/fl}$ mice 30 and 75 days after intradermal paw injection with PBS or Diphtheria toxin. 70 kDa Rh-D was injected into the paws and the transport of the molecule was monitored by fluorescent microscopy 90 min post Rh-D administration. Quantitative data are represented as mean and SEM in 5–6 popliteal lymph nodes in each group (two-tailed, paired $T$-test, $P = 4.51 \times 10^{-5}$ after 30 days for 6 mice and $P = 0.0421$ after 75 days for 5 mice). Asterisks indicate $P < 0.05$ compared with control.

**Digestion of ear samples and flow cytometry analysis**. Ear samples from *Prox1$^{GFP}$* and C57Bl/6 wild-type mice were collected and cut into small pieces, then digested with the Liberase II kit (Roche) with Eppendorf Thermomixer at 1400 rpm for 1 h at 37 °C[61]. Thereafter a 70-μm cell strainer was used (BD). The number of LECs and infiltrating leukocytes was tested by a BD Biosciences FACSCalibur cytometer (data acquisition was performed using BD CellQuest Pro (Becton Dickinson, version: 6.1)). Lymphatic endothelial cells were shown by GFP signal and immunostaining with PE anti-mouse LYVE1 (R&D Systems, FAB2125P) in a dilution of 1:200. Leukocytes were shown by immunostaining with the following antibodies in a dilution of 1:200: Alexa Fluor 488 anti-mouse CD206 (BioLegend,

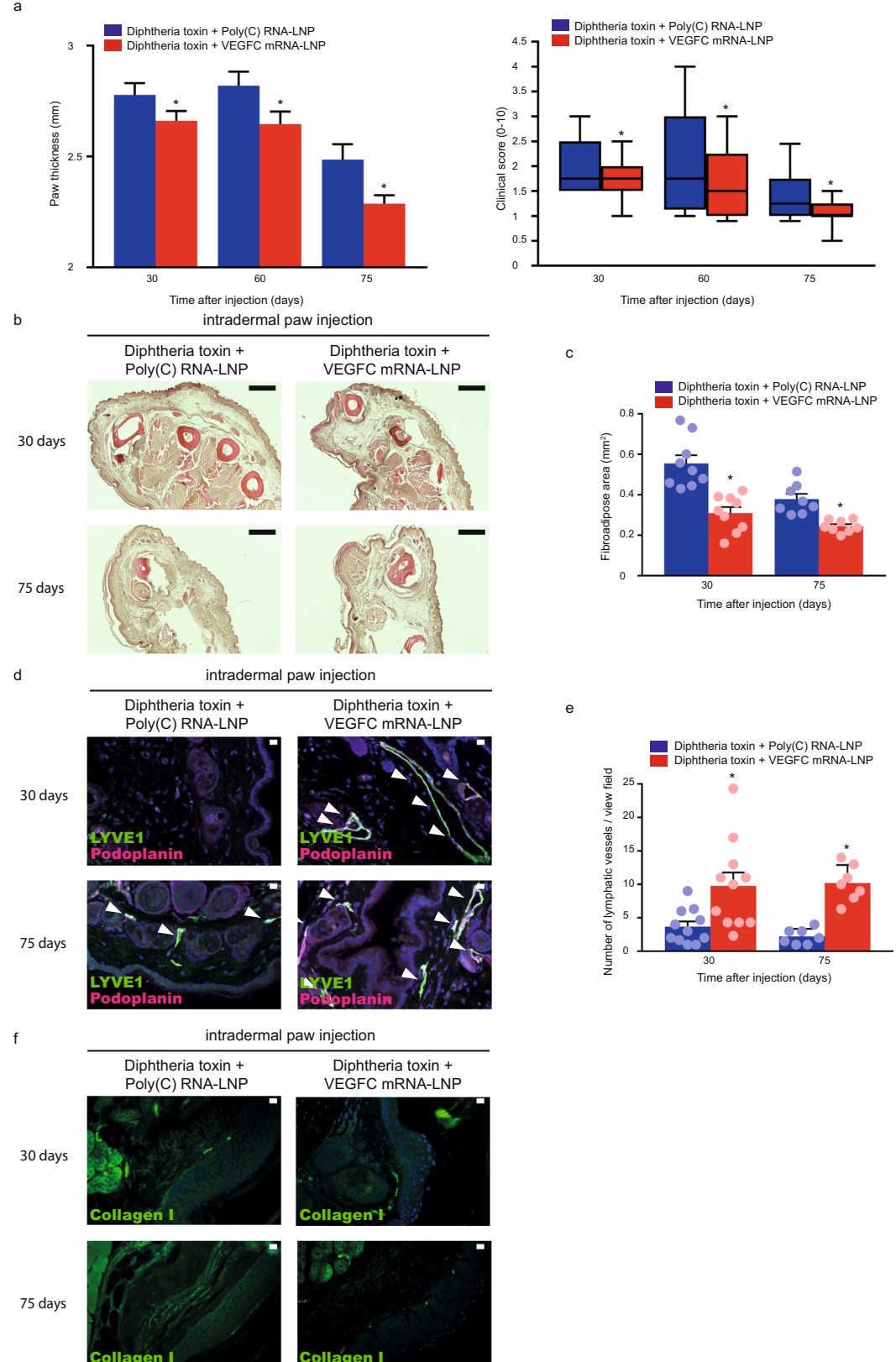

BZ-141710), Alexa Fluor 647 Rat Anti-Mouse CD3 (BD Biosciences, 557869), PE Rat Anti-Mouse CD45R/B220 (BD Biosciences, 553090), PerCP-Cy5.5 anti-mouse CD11b (BD Biosciences, 550990) PE anti-mouse CD45 (Biolegend, 103105, Clone: 30-F11), FITC anti-mouse Ly6G/C Antibody (BD Biosciences, 553127, Clone: RB6-8C5)[60,61]. In parallel, 1, 5, 10, 15, and 20 days after 1 μg of Poly(C) or VEGFC

mRNA-LNP injection, ear samples were harvested and digested with Liberase II kit (Roche). In vivo VEGFC expression of the interstitial fluid of these ear samples was analyzed by Western blot analysis and ELISA. Examples of gating strategy are shown in the Supplementary Information file (Supplementary Fig. 8). All investigators performing flow cytometry were blinded for the sample origin.

**Fig. 9 VEGFC mRNA-LNP treatment reverses experimental lymphedema. a** Monitoring paw thickness and clinical score in Diphtheria toxin and tamoxifen-treated Flt4-CreER$^{T2}$; *iDTR$^{fl/fl}$* mice intradermally injected with 1 μg of Poly(C) or VEGFC mRNA-LNPs. Quantitative data are represented as mean and SEM for paw thickness and box shows median and 25$^{th}$ to 75$^{th}$ percentiles and whiskers show 10$^{th}$–90$^{th}$ percentiles for clinical score in 15–31 mouse paws in each group (two-tailed, paired *T*-test, for paw thickness $P = 0.0090$ on days 30 for 31 mice, $P = 0.0053$ on day 60 for 25 mice and $P = 0.0082$ on day 75 for 15 mice. Two-tailed Wilcoxon signed-rank test for clinical score $P = 0.0050$ on day 30 for 31 mice, $P = 0.0278$ on day 60 for 25 mice, and $P = 0.0469$ on day 75 for 15 mice). **b** Haematoxylin and Eosin histology of the paws of tamoxifen-treated Flt4-CreER$^{T2}$; *iDTR$^{fl/fl}$* mice 30 and 75 days after treatment with Diphtheria toxin and 1 μg of Poly(C) or Diphtheria toxin and 1 μg of VEGFC mRNA-LNPs intradermally. Poly(C) or VEGFC mRNA-LNPs were injected 8 days after Diphtheria toxin treatment. Representative images are shown of 5 mouse paws in each group (bars, 200 μm). **c** Paw fibroadipose area of tamoxifen-treated Flt4-CreER$^{T2}$; *iDTR$^{fl/fl}$* mice 30 and 75 days after treatment with Diphtheria toxin and 1 μg of Poly(C) or Diphtheria toxin and 1 μg of VEGFC mRNA-LNPs. Poly(C) or VEGFC mRNA-LNPs were injected 8 days after Diphtheria toxin treatment. Quantitative data are represented as mean and SEM in 8–9 mouse paws in each group stained with Haematoxylin and Eosin (two-tailed, paired *T*-test, $P = 0.0001$ on day 30 for 9 mice and $P = 0.0047$ on day 75 for 8 mice). **d** Anti-LYVE1, anti-Podoplanin immunofluorescent histology of the paws of tamoxifen-treated Flt4-CreER$^{T2}$; *iDTR$^{fl/fl}$* mice 30 and 75 days after treatment with Diphtheria toxin and 1 μg of Poly(C) or Diphtheria toxin and 1 μg of VEGFC mRNA-LNPs. Poly(C) or VEGFC mRNA-LNPs were injected 8 days after Diphtheria toxin treatment. Representative images are shown of 5 mouse paws in each group (bars, 50 μm). Arrows indicate LYVE1 and Podoplanin double positive lymphatic vessels. **e** Number of lymphatic vessels of tamoxifen-treated Flt4-CreER$^{T2}$; *iDTR$^{fl/fl}$* mice 30 and 75 days after the paw treatment with Diphtheria toxin and 1 μg of Poly(C) or Diphtheria toxin and 1 μg of VEGFC mRNA-LNPs. Poly(C) or VEGFC mRNA-LNPs were injected 8 days after Diphtheria toxin treatment. Quantitative data are represented as mean and SEM in 7–11 mouse paws in each group (two-tailed, paired *T*-test, $P = 0.0024$ after 30 days for 11 mice and $P = 0.0008$ after 75 days for 7 mice). **f** Anti-Collagen I immunofluorescent histology of the paws of tamoxifen-treated Flt4-CreER$^{T2}$; *iDTR$^{fl/fl}$* mice 30 and 75 days after treatment with Diphtheria toxin and 1 μg of Poly(C) or Diphtheria toxin and 1 μg of VEGFC mRNA-LNPs. Poly(C) or VEGFC mRNA-LNPs were injected 8 days after Diphtheria toxin treatment. Representative images are shown of 5 mouse paws in each group (bars, 50 μm). Asterisks indicate $P < 0.05$ compared with control. All cell nuclei are labeled with DAPI (blue) in paraffin-embedded tissues.

**Histological procedures and immunostaining**. Isolated tissues were fixed in 4% paraformaldehyde (Sigma-Aldrich) overnight, dehydrated in 50, 70, 95, and 100% ethanol, and embedded in paraffin using a Leica EG1150H embedding station. 8-μm-thick sections using a Thermo Scientific microtome (HM340E) were processed for Haematoxylin and Eosin (Leica) staining and different type of immunostainings. The immunostainings were performed by using the following antibodies anti-LYVE1 (R&D systems, AF2125), anti-CD31 (R&D systems, mab3628), anti-von Willebrand Factor (anti-vWF) (Sigma-Aldrich, F3520), anti-Podoplanin (R&D systems, AF3244 and Abcam, ab92319), anti-α-SMA (Abcam, ab124964), anti-Collagen I antibody (Abcam, ab34710), and anti-Ly6G/C (anti-GR1) (BD Pharmingen, 550291, clone RB6-8C5) in a dilution of 1:100. Anti-goat, anti-rabbit, anti-hamster, and anti-rat secondary antibodies were used conjugated to Alexa Fluor 488, 568, and 594 (Life Technologies, anti-goat AF488: A11055, AF568: A11057, anti-rat AF568: A11077, anti-hamster AF594: A21113, anti-mouse AF488: A11029, anti-mouse AF568: A11031, anti-mouse AF594: A32744, anti-rabbit AF488: AF11034, anti-rabbit AF568 AF11036) in a dilution of 1:250. 4′,6-Diamidino-2-phenylidole (DAPI) staining was used to determine the number of nuclei and to assess gross tissue morphology in paraffin-embedded tissues. Microscopic images were taken by an ECLIPSE Ni-U microscope (Nikon Instruments) connected to Nikon DS-Ri2 camera.

**Monitoring the proliferation of the lymphatic vessels**. To monitor the proliferation of the lymphatic vessels 50 mg/kg EdU was injected intraperitoneally 4 days after 1 μg of Poly(C) RNA-LNP or VEGFC mRNA-LNP treatment of the ear of mice. The samples were harvested 24 h after EdU treatment, and histological procedures were performed. The slides were stained with Click-iT EdU Cell Proliferation Kit for Imaging, Alexa Fluor 594 dye (Invitrogen, C10337), anti-LYVE1 antibody (R&D systems, AF2125), and anti-α-SMA antibody (Abcam, ab124964). Confocal images were acquired using a Nikon Eclipse A1 Confocal Laser Microscope (Nikon Instruments) using 10x or 40x dry objectives.

**Monitoring lymphatic function in vivo**. To test lymphatic function, intradermal injection of 0.5 μl (ears) or 2.5 μl (paws) of 70 kDa tetramethylrhodamine-dextran (Rh-D) (Life Technologies) at 10 mg/ml using a 34 G needle connected to a 5 μl glass syringe (VWR International) was performed into previously 1 μg of Poly(C) RNA-LNP or VEGFC mRNA-LNP treated ears or paws of *Prox1$^{GFP}$* lymphatic reporter and wild-type animals 22 days by ears and 75 days by paws after the treatment, as described[4,9,10]. In tamoxifen-treated PBS or Diphtheria toxin injected FLT4-CreER$^{T2}$; *iDTR$^{fl/fl}$* mice 30 or 75 days after the first Diphtheria toxin injection and tamoxifen and both hind paws Diphtheria toxin treated FLT4-CreER$^{T2}$; *iDTR$^{fl/fl}$* mice 30 or 75 days post 1 μg of Poly(C) and VEGFC mRNA-LNP treatment were also injected with 2.5 μl of 70 kDa Rh-D (Life Technologies) at 10 mg/ml. The contralateral ears or paws of the same animal were used for Poly(C) and VEGFC mRNA-LNP treatment. Selective uptake and passage of Rh-D was monitored by a fluorescent research stereo microscope (Nikon SMZ25) equipped with a Nikon DS-Ri2 camera in ears and hind limbs. Images are shown at 60 (ear) or 90 (hind limb) minutes after the tracer injection. In paw injected animals the fluorescent signal was also monitored in the popliteal lymph nodes. Investigators performing experiments monitoring lymphatic function in vivo were blinded for the treatment of the mice.

**Quantification of microscopic images**. NIH Fiji software[62] (versions 1.52i -1.53 f) and NIS-Elements Imaging Software (Nikon, version BR 4.60.00) programs were used for the quantification of microscopy image-based data. The length, diameter, and branching point number of lymphatics were quantified in a standardized area on stereo microscopic images in each ear and back skin of each animal. Length of the lymphatic network was measured by drawing over lymphatic vessel specific signals with the freehand selection tool and the overall length of selection was registered. Lymphatic vessel diameters were measured with the straight segment tool on 20 different spots. Branching points were counted in a standardized area of each image. Mitotic index was calculated as the ratio of EdU positive and all lymphatic endothelial cell nuclei per view field in each ear. Blood vessel number was counted as vWF high and LYVE1 negative vessel-like structures per view field of 3 images per histological slide in each ear. Fibroadipose area were measured in Haematoxylin and Eosin stained slides in a standardized area marked with the freehand selection tool in each hind limb of each animal. Lymphatic vessel number was counted as LYVE1 and Podoplanin double positive vessel-like structures per view field of 3 images per histological slide in each hind limb of each mouse. In vivo lymphatic function (Figs. 6, 8, and 10) was determined by measuring the mean fluorescent intensity of popliteal lymph nodes of previously 70 kDa Rh-D paw injected animals and the mean fluorescent intensity of three equal area (ROI) in the background was subtracted in each hind limb of each animal. Investigators performing microscopic imaging and quantification of microscopic images were blinded for treatment and sample origin.

**Presentation of data and statistical analysis**. The number of the used animals and independent experiments are labeled in the Figure legends. All macroscopic pictures and microscopic images are representative of the indicated number of independent experiments. Individual data points with mean and SEM or median and interquartile range (IQR) are shown in each diagram. Box plots are shown with whiskers of 10–90 percentiles. Microscopic image processing and analysis were performed using NIS-Elements Imaging Software (Nikon, version BR 4.60.00), Photoshop CS6 (Adobe, version 16.0.0), and NIH Fiji software (versions 1.52i -1.53 f). Flow cytometry result processing and analysis were performed using FCS Express 6 Flow Cytometry Software (De Novo Software, version 6.06.0033. Data processing and statistical analyses were performed using Graphpad Prism (version 7.03) and Excel 2018 software. All datasets were tested for normality using Shapiro–Wilk test of normality. A dataset was considered to be normally distributed with $P > 0.01$. Normally distributed datasets were tested with Student paired or unpaired two-tailed *T*-test and non-normally distributed datasets were tested with Wilcoxon signed-rank test. A difference was considered statistically significant at $P < 0.05$.

**Reporting summary**. Further information on research design is available in the Nature Research Reporting Summary linked to this article.

## Data availability
All the data supporting the findings of this study are available within the article and its supplementary information files and from the corresponding authors upon reasonable

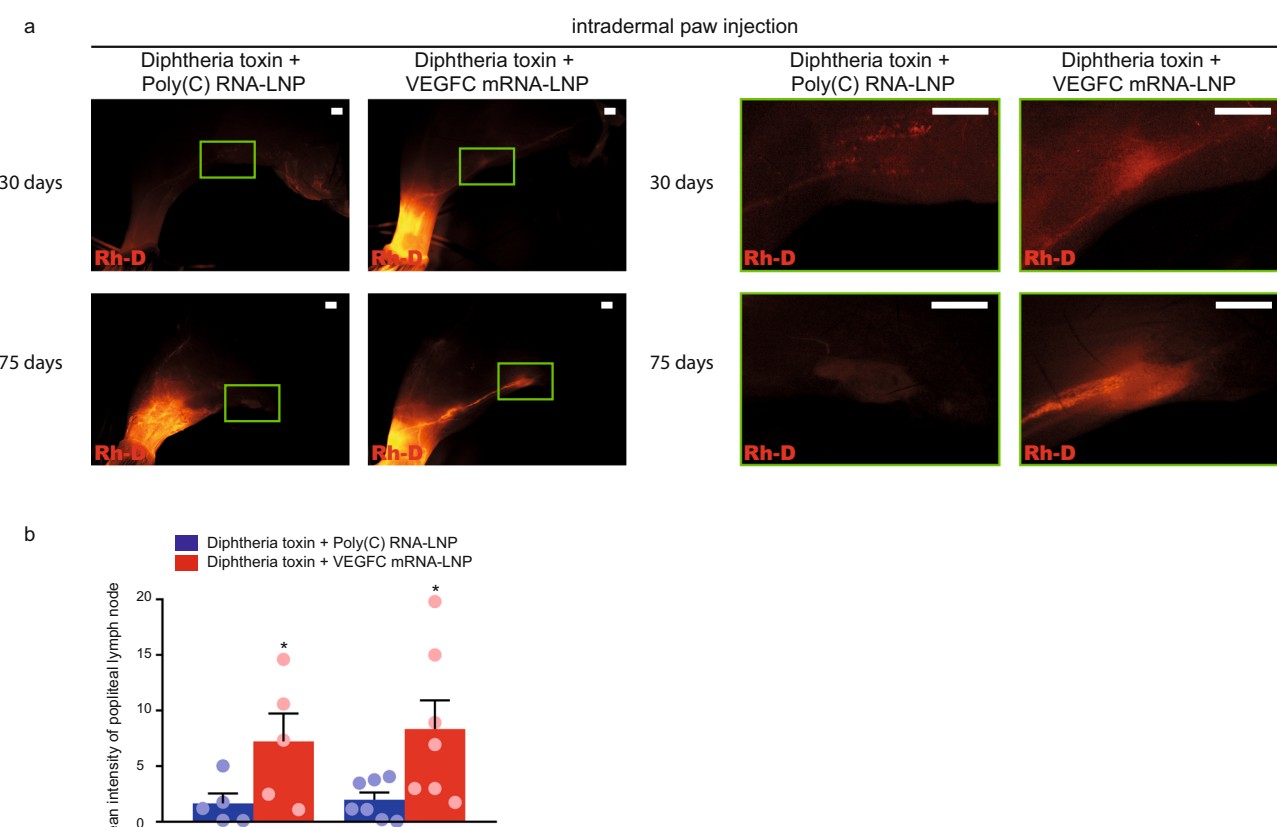

**Fig. 10 VEGFC mRNA-LNP treatment restores lymphatic function in experimental lymphedema. a** Monitoring active lymphatic function in the contralateral hind limbs of the same tamoxifen-treated Flt4-CreER$^{T2}$; iDTR$^{fl/fl}$ mouse 30 or 75 days after the injection of Diphtheria toxin and 1 µg of Poly (C) into one paw and Diphtheria toxin and 1 µg of VEGFC mRNA-LNPs into another paw. Poly(C) or VEGFC mRNA-LNPs were injected 8 days after Diphtheria toxin treatment. 70 kDa Rh-D was injected into the paws and the transport of the molecule was monitored by fluorescent microscopy 90 min post Rh-D administration. Representative images are shown of 5-7 mouse hind limbs in each group (bars, 1000 µm). Green rectangles show the magnified area which represent the area of popliteal lymph nodes. **b** Fluorescent intensity of popliteal lymph nodes of tamoxifen-treated Flt4-CreER$^{T2}$; iDTR$^{fl/fl}$ mice 30 and 75 days after the injection of Diphtheria toxin and 1 µg of Poly(C) into one paw and Diphtheria toxin and 1 µg of VEGFC mRNA-LNPs into another paw. Poly(C) or VEGFC mRNA-LNPs were injected 8 days after Diphtheria toxin treatment. 70 kDa Rh-D was injected into the paws and the transport of the molecule was monitored by fluorescent microscopy 90 min post Rh-D administration. Quantitative data are represented as mean and SEM in 5–7 popliteal lymph nodes in each group (two-tailed, paired $T$-test, $P = 0.0425$ after 30 days for 5 mice and $P = 0.0373$ after 75 days for 7 mice). Asterisks indicate $P < 0.05$ compared with control.

request. A reporting summary for this article is available as a Supplementary Information file. Source data are provided with this paper.

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

## Acknowledgements

We thank V. Németh, E. Marinkás, and D. Csengel for excellent technical assistance, B. Enyedi and M. Lengyel for help with experiments and Young-Kwon Hong for *Prox1^GFP* mice. This work was supported by the Lendület program of the Hungarian Academy of Sciences (LP2014-4/2019 to Z. Jakus), the National Research, Development and Innovation Office (NVKP_16-2016-1-0039 to Z. Jakus), the European Union and the Hungarian Government (VEKOP-2.3.2-16-2016-00002 to Z. Jakus and EFOP-3.6.3-VEKOP-16-2017-00009 to D. Szőke and G. Kovács), and the Higher Education Institutional Excellence Program of the Ministry for Innovation and Technology in Hungary, within the framework of the Molecular Biology thematic program of the Semmelweis University. D. Weissman and N. Pardi were supported by the NIAID of the NIH under award numbers R01-AI050484, R01-AI124429, R01-AI084860, and R01-AI146101.

## Author contributions

D.Sz., N.P., and Z.J. designed the work, interpreted the results, and wrote the paper. D.Sz., G.K., É.K., L.B., K.Sz-A., P.A., A.S.D., and Z.J. performed the experiments and analyzed the data. N.P. designed and produced the nucleoside-modified mRNAs. B.L.M., Y.K.T., T.D.M., and M.J.H. developed and prepared the lipid nanoparticles. B.J.M and R.P.K. developed and provided the genetic secondary lymphedema model. K.K. and D.W. provided experimental tools. Z.J. supervised the project.

## Competing interests

In accordance with the University of Pennsylvania policies and procedures and our ethical obligations as researchers, we report that Katalin Karikó and Drew Weissman are named on patents that describe the use of nucleoside-modified mRNA as a platform to deliver therapeutic proteins. Drew Weissman and Norbert Pardi are also named on a patent describing the use of modified mRNA in lipid nanoparticles. We have disclosed those interests fully to the University of Pennsylvania, and we have in place an approved plan for managing any potential conflicts arising from licensing of our patents. No competing interests from other authors have been reported.

**Additional information**

