## [Peer Review File · Nature Communications]

Reviewers' Comments:

Reviewer #1:

Remarks to the Author:

In this manuscript, the authors utilized the IVT mRNA encapsulated lipid nanoparticles encoding murine VEGFC to stimulate the lymphatic growth and reduce lymphedema in vivo. Their VEGFC mRNA-LNPs demonstrated the ability to induce specific lymphatic growth and this effect lasted up to 60 days after the administration of a single low dose of LNPs. The LNPs also reversed edema formation in the lymphedema animal model without obvious adverse effects. The manuscript is well-written and the results are informative. Several minor concerns are listed below:

1. In Figures 1-3 images, what's the blue stains representing? In Figure 1h, why were blue stains not shown in the immunofluorescence section of lung tissues?
2. In Figure 3c, the authors investigated the lymphatic and blood vessel proliferation after intradermal ear injection. Did they imaged CD31 and GFP signal after intradermal back skin injection?
3. In page 7, line 3, the authors mentioned that LNPs treatment only caused modest increase of CD45 positive immune cells. Since the number of CD45 positive immune cells is significantly higher than control and the significance level is the same as GFP study ($p < 0.05$), why did the authors state it was the "modest increase"? How did the authors define the "modest"?
4. In page 7, line 18, the authors concluded that the formation of new lymphatic vessels was fully functional. It is hard to see the difference from figure 4a. Also, there's no images to compare the lymphatic vascular morphology before and after LNP treatment. Please clarify the statement.
5. In the last section of the results, the authors used "clinical score" to evaluate the signs of lymphedema. Please provide specific standard of this evaluation criteria in the method.

Reviewer #2:

Remarks to the Author:

In the manuscript, the authors claimed that the nucleoside-modified mRNA-LNP is a novel therapeutic platform to deliver protein by demonstrating the therapeutic effects of VEGFC mRNA-LNPs in experimental lymphedema through VEGFC-induced lymphangiogenesis. Although the approach is technologically meritorious, there is no specific novelty in this study. Furthermore, many data are just descriptive and are not convincing to support the conclusion

1. Overall, data images for the growth of lymphatic vessels in vivo are insufficient. Most of the immunohistochemistry data were not due to their strong background and low resolutions (e.g. Fig. 1c, 1h, 2d, 3a-c, 4a, Suppl. Fig. 2a and b). For example in 1C, the background between the control and VEGFC treated pictures are quite different (one is dark and the other is bright).
2. Another important problems of this study is that the shown pictures have different morphologies of lymphatic vessels such as shown in Figure 1b between control and VEGFC. In the control group (Poly(C) RNA-LNP), the cross sectional image does not show any LYVE1-positive vessels
3. In Figure 2a and b, the authors demonstrated that mRNA-LNP dose not have off-target effects in organs without mRNA-LNP using GFP mRNA-LNP. However, data analysis in four hours after injection was inappropriate to determine whether mRNA-LNP have off-target effects or not. The authors should conduct analysis of lymphatic networks in various time points and later than 4 hours post-injection of VEGFC mRNA-LNP.
4. In Figure 2, the authors mentioned organ-specific lymphatic growth by VEGFC mRNA-LNP. However, the data in Figure 2 and description in the Legends and Result sections were inconsistent. There was no accurate conclusion made by the data and authors. The authors should correct the legend, the labeling in the data figure, and describe the results including conclusion regarding Figure 2 in the Result section. (e.g. in Fig. 2c, VEGFC mRNA-LNP was injected into ear only, or three organs including ear, lung, and small intestine?)
5. In Figure 4, the authors showed therapeutic effects of VEGFC mRNA-LNPs using a genetic secondary lymphedema model (FLT4-CreERT2; iDTRfl/fl mice). The experimental model is innovative, but data were not sufficient to support the author's conclusion on the therapeutic

effects of VEGFC mRNA LNPs in the lymphedema model. Detailed experimental design also should be provided in the manuscript based on the paper published by Gardenier et al., in 2016, JCI Insight. (e.g. measurement of foot diameter, fibroadipose area, lymphatic capillary density, and lymphatic function using NIR lymphoscintigraphy). Furthermore, the authors should provide supplementary data to address whether the experimental lymphedema model is established well.

6. In addition, the time point of measuring lymphedema is quite wrong. Lymphedema is usually a chronic process so the end point should be at least several weeks or months later to compare the secondary changes in dermis and subcutaneous tissues.

7. No western blot was performed to confirm the protein production of VEGFC-mRNA-LNP in vivo

8. In Figure 4i, in the Diphtheria toxin model, LYVE1 is still expressed, which is different from the description of the authors. Given the difference of DAPI expression between PBS and Diph toxin groups, the low staining of LYVE1 is due to the difference in imaging or staining quality. Thus it is not clear whether their genetic secondary lymphedema model is appropriate.

9. There is no direct evidence in vivo that VEGFC-mRNA-LNP is really proliferating lymphatic vessels. This should be performed by double-staining with a lymphatic marker and a proliferation marker such as BrdU or Ki67.

10. No quantification data were provided in lymphatic vessels data measured in histology. Most of the conclusions were made by the representative images.

Other minor points include:

1. Please show VEGFC expression levels after transfection of VEGFC-mRNA LNPs into HEK293 cells at various time points in Figure 1a.

Reviewer #3:

Remarks to the Author:

This manuscript explores the potential of nucleoside-modified RNA encoding VEGFC to promote growth of lymphatic vessels and reverse experimental lymphoedema in mice. Lymphangiogenic proteins, such as VEGF-C, have been used many times in the past for attempts to achieve this, but with mixed results. The novelty here is not the pro-lymphangiogenic strategy but the use of nucleoside modified mRNA as the delivery vehicle. Overall the results are clearly documented and impressive. However the manuscript is relatively silent on the nature of the changes to lymphatic vessels. My specific questions are outlined below.

1. What is the nature of the change in lymphatics induced by the nucleoside-modified RNA encoding VEGFC? For example, Figure 1d-g shows graphs about "normalized length of the lymphatic network". What does this mean? Does it mean the average length of lymphatics? This is a key to understanding how the RNA elicits its effects. Were the lymphatics of wider caliber? What type of lymphatic vessel was influenced by the RNA? Are there more branch points? How is the morphology altered? A more thorough morphological analysis is required.

2. In Figure 2 the Authors explore the systemic effects of their approach. But they use only 0.2 microgram which is a lower dose than in other parts of the study. And they examine the effects in lung and small intestine after only 4 hours. Why not use the 1 microgram dose and give more time for effects on lymphatics at distant sites to become apparent? This seems not to be a sufficiently rigorous assessment of systemic effects.

Ms. Ref. No.: NCOMMS-19-01080A (Revision)

Title: Nucleoside-Modified VEGFC mRNA Induces Organ-Specific Lymphatic Growth and Reverses Experimental Lymphedema

We would like to thank the reviewers for their thoughtful comments and questions. We have performed a large number of new experiments to address their specific concerns. We believe that we have fully addressed all their points in a highly rigorous and definitive fashion. Importantly, the inclusion of all the additional experiments suggested by the reviewers has significantly improved the manuscript. In particular, we have added 3 new figures, 3 new supplementary figures, 18 brand new figure panels, replaced 10 panels with new representative images, updated 10 panels with additional new experimental data, included the results of several additional experiments and statistical calculations in the text. The major revisions to the manuscript that address these points include the following:

- 1) We rigorously evaluated the long-term effects of nucleoside-modified VEGFC mRNA treatment. After detailed validation of the genetic model, our new results demonstrate that VEGFC mRNA-LNP treatment reverses experimental secondary lymphedema by restoring lymphatic function (Figs. 6 and 7).
- 2) We have included additional experiments to monitor the organ-specific effects of treatment with VEGFC mRNA-LNPs. In addition to the visualization of lymphatic morphology in paraffin-embedded tissues and whole-mount organs, we have also performed flow cytometry to assess organ-specificity (Fig. 3).
- 3) We rigorously analyzed immune cell infiltration after VEGFC mRNA-LNP treatment using several immune cell markers (Figs. 4d-e).
- 4) We have performed ELISA and immunoblot experiments to demonstrate VEGFC secretion into the interstitial fluid *in vivo* induced by the VEGFC mRNA-LNP platform (Figs. 1b-c).
- 5) We have performed experiments to monitor the proliferation of lymphatic endothelial cells based on their EdU incorporation (Figs. 2b-c).
- 6) The quality of several representative images has been improved and additional markers have been used to identify the various cell types in most of the panels.
- 7) Detailed quantification of the experimental data has been included in the revised manuscript.

Most importantly, all the new experiments confirmed our original conclusions that 1) describe a novel *in vivo* application of the nucleoside-modified mRNA-LNP therapeutic protein delivery platform, 2) describe an approach for identifying the organ-specific physiological and pathophysiological roles of the lymphatic system, and importantly 3) propose an efficient and safe treatment option that may serve as the basis for the development of a novel therapeutic tool to cure or reduce lymphedema. Our study describes the first use of the nucleoside-modified mRNA in the field of lymphatics, and based on the data presented in the manuscript we believe that the use of this platform is a viable strategy for the treatment of secondary lymphedema.

Our responses to the specific points are provided below. Corresponding changes in the manuscript are highlighted by red font color. We hope that our study is now acceptable for publication in Nature Communications.

Reviewers' comments:

Reviewer #1

Reviewer's comment: In this manuscript, the authors utilized the IVT mRNA encapsulated lipid nanoparticles encoding murine VEGFC to stimulate the lymphatic growth and reduce lymphedema in vivo. Their VEGFC mRNA-LNPs demonstrated the ability to induce specific lymphatic growth and this effect lasted up to 60 days after the administration of a single low dose of LNPs. The LNPs also reversed edema formation in the lymphedema animal model without obvious adverse effects. The manuscript is well-written and the results are informative. Several minor concerns are listed below: 1. In Figures 1-3 images, what's the blue stains representing? In Figure 1h, why were blue stains not shown in the immunofluorescence section of lung tissues?

We thank the Reviewer for the kind comments. The blue signal in our paraffin-embedded histology slides is a DAPI staining to label and visualize the nuclei of the cells. The DAPI signal is shown to indicate the gross morphology of the immunostained organs. In the revised manuscript the presence of the DAPI staining in the histology slides is appropriately indicated. The DAPI staining of the lung tissue was also included to make the manuscript more consistent as suggested by the Reviewer (Fig. 2g).

Reviewer's comment: 2. In Figure 3c, the authors investigated the lymphatic and blood vessel proliferation after intradermal ear injection. Did they imaged CD31 and GFP signal after intradermal back skin injection?

In our original Figure 3c the CD31 staining and *Prox1*^{GFP} signal were shown in whole-mount ear samples after Poly(C) and VEGFC mRNA-LNP treatments. We realized that the quality of the staining was not sufficient, therefore, we replaced the panel in the revised Fig. 4b. In addition, in our revised submission we have quantified the number of blood vessels in Fig. 4c based on the vWF staining and the absence of lymphatic marker expression. Our new data support our original conclusion indicating that VEGFC mRNA-LNP treatment induces strong lymphatic vessel proliferation, while does not affect significantly the growth of the blood vasculature.

Reviewer's comment: 3. In page 7, line 3, the authors mentioned that LNPs treatment only caused modest increase of CD45 positive immune cells. Since the number of CD45 positive immune cells is significantly higher than control and the significance level is the same as GFP study ($p < 0.05$), why did the authors state it was the "modest increase"? How did the authors define the "modest"?

We thank the reviewer for this useful comment. We have performed a series of new experiments to monitor the possible infiltration of various immune cells after Poly(C) RNA-LNP and VEGFC mRNA-LNP injections. In the revised manuscript we show that our platform does not induce local inflammation by monitoring pan-leukocyte (CD45), granulocyte (Ly6GC), monocyte (Ly6GC negative and CD11b positive), T cell (CD3), B cell (B220) and CD206 markers by flow cytometry and immunostaining of histology slides (Figs. 4d-e). None of these immune cell markers showed a significant change in the VEGFC mRNA-LNP-injected organs.

Reviewer's comment: 4. In page 7, line 18, the authors concluded that the formation of new lymphatic vessels was fully functional. It is hard to see the difference from figure 4a. Also, there's no images to compare the lymphatic vascular morphology before and after LNP treatment. Please clarify the statement.

As suggested by the Reviewer, we have provided a more detailed analysis and discussion of the experiments in which lymphatic function was monitored after control and VEGFC mRNA-LNP treatments. To strengthen this data, we also performed additional experiments to demonstrate that the new lymphatic vessels are fully functional. In addition to monitoring the lymphatic function in the mouse ear, we have set up a new system to follow the lymphatic function in the hind limb. Our results indicate that lymphatic vessels take up and transport large molecular weight molecules after control treatment in both the ear and hind limb (Figs. 5a-b). Importantly, increased number of lymphatic vessels can be detected in the VEGFC mRNA-LNP injected organs, and this dense network of lymphatic vessels also takes up and transports the 70 kDa Rhodamine-dextran without signs of significant leakage of the macromolecule (Figs. 5a-b). The transport of the labelled macromolecule can also be detected to the lymph nodes (specifically to the popliteal lymph node) (Fig. 5b).

Reviewer's comment: 5. In the last section of the results, the authors used "clinical score" to evaluate the signs of lymphedema. Please provide specific standard of this evaluation criteria in the method.

In addition to the thickness measurements, which is an objective parameter, we have set up a clinical score system as a subjective parameter to assess lymphedema severity following the injection of PBS and Diphtheria toxin, and in parallel post Poly(C) and VEGFC mRNA-LNP treatment. The clinical score system allows the more detailed assessment of the clinical disease indicators. Therefore, visible clinical signs of secondary lymphedema were scored on a 0-10 scale by two independent investigators blinded for the treatment of the mice. The evaluation criteria have been added to the methods section of the revised manuscript. The paws were scored based on their swelling (0-4), thickness (0-2), color (0-2) and skin tightness (0-2).

Reviewer #2

Reviewer's comment: In the manuscript, the authors claimed that the nucleoside-modified mRNA-LNP is a novel therapeutic platform to deliver protein by demonstrating the therapeutic effects of VEGFC mRNA-LNPs in experimental lymphedema through VEGFC-induced lymphangiogenesis. Although the approach is technologically meritorious, there is no specific novelty in this study. Furthermore, many data are just descriptive and are not convincing to support the conclusion.

In vitro transcribed nucleoside-modified mRNA-LNP emerged as an easy-to-produce, potent and safe therapeutic delivery platform for vaccine development, protein replacement therapy and in vivo genome editing (reviewed in Sahin et. al., Nature Reviews Drug Discovery, 2014, PMID: 25233993 and Pardi et. al., Nature Reviews Drug Discovery, 2018, PMID: 29326426). Our study describes the first use of the nucleoside-modified mRNA in the field of lymphatics, and based on the data presented in the manuscript we believe that the use of this platform is a viable strategy for the treatment of secondary lymphedema. We demonstrated that

this platform works just as well as the adenovirus or AAV-based systems (described and evaluated by others) but represents a much safer approach for disease treatment. Thus, we respectfully disagree with the Reviewer and strongly feel that there are a number of critically important novel aspects of the experiments described within this manuscript: Our current studies 1) describe a novel *in vivo* application of the nucleoside-modified mRNA-LNP therapeutic protein delivery platform, 2) describe an approach for identifying the organ-specific physiological and pathophysiological roles of the lymphatic system, and importantly 3) propose an efficient and safe treatment option that may serve as the basis for the development of a novel therapeutic tool to cure or reduce lymphedema. We tried to more clearly emphasize the above-mentioned novel aspects of our work in the revised manuscript.

In addition, we have performed a large number of new experiments to address the concerns and questions raised by the Reviewer and we indeed feel that the additional experiments helped to expand the project and has significantly improved the manuscript and our new additions support the final conclusions.

Reviewer's comment: 1. Overall, data images for the growth of lymphatic vessels in vivo are insufficient. Most of the immunohistochemistry data were not due to their strong background and low resolutions (e.g. Fig. 1c, 1h, 2d, 3a-c, 4a, Suppl. Fig. 2a and b). For example in 1C, the background between the control and VEGFC treated pictures are quite different (one is dark and the other is bright).

We completely agree with the Reviewer that some of our original images failed to provide a high quality assessment of lymphatic growth *in vivo*. Therefore, we have replaced several photos in the representative panels. In addition, in most of the panels we use double staining to identify particular cell types. We also apologize for the low resolution of the submitted file. This is due to the need to compress the file to a single PDF file. We have attempted to do this with improved resolution the second time. It should be noted that the primary data are of higher resolution.

Reviewer's comment: 2. Another important problems of this study is that the shown pictures have different morphologies of lymphatic vessels such as shown in Figure 1b between control and VEGFC. In the control group (Poly(C) RNA-LNP), the cross sectional image does not show any LYVE1-positive vessels.

We thank the Reviewer for this useful comment. We agree, the representative image which was shown in the original Fig. 1b panel for the Poly(C) RNA-LNP treated ear was not convincing because of the absence of a lymphatic structure. As suggested, we have replaced this and many other representative images. To further strengthen the revised manuscript, we show the morphology of the lymphatic vessels at least by two separate markers in most of the panels. Our results indicate that lymphatic vessels are present in the control treated ear and VEGFC mRNA-LNP treatment induces the proliferation of lymphatic vessels of the mouse ear as it is shown by representative images (Fig. 2a) and quantified data (Figs. 2d-f).

Reviewer's comment: 3. In Figure 2a and b, the authors demonstrated that mRNA-LNP dose not have off-target effects in organs without mRNA-LNP using GFP mRNA-LNP. However, data analysis in four hours after injection was inappropriate to determine whether mRNA-LNP have

off-target effects or not. The authors should conduct analysis of lymphatic networks in various time points and later than 4 hours post-injection of VEGFC mRNA-LNP.

We completely agree with the Reviewer. We have clarified this part of the manuscript. In these experiments 1 μg GFP mRNA-LNP was injected into the back skin (or ear in separate animals) and GFP expression was monitored in the ear, lung and small intestine 4 hours after the injection (Suppl. Fig. 4). Our results indicate that strong local GFP expression can be detected at the injection site, but not in other organs including the lung and the small intestine in the same animal. These results are in agreement with our previous report (Pardi et. al, Journal of Controlled Release, 2015, PMID: 26264835), in which the different delivery routes of the mRNA-LNP platform were compared in a luciferase reporter assay *in vivo*. Of note, a possible systemic effect is detectable at 4 hours as it is indicated in the latter study. We moved the GFP mRNA-LNP data to the supplementary material (Suppl. Fig. 4) in the revised manuscript to separate it from the experiments characterizing the VEGFC mRNA-LNP platform.

Reviewer's comment: 4. In Figure 2, the authors mentioned organ-specific lymphatic growth by VEGFC mRNA-LNP. However, the data in Figure 2 and description in the Legends and Result sections were inconsistent. There was no accurate conclusion made by the data and authors. The authors should correct the legend, the labeling in the data figure, and describe the results including conclusion regarding Figure 2 in the Result section. (e.g. in Fig. 2c, VEGFC mRNA-LNP was injected into ear only, or three organs including ear, lung, and small intestine?)

We agree with the Reviewer and we have revised this part of the manuscript. In addition to the analysis of lymphatic growth by upright microscopy on paraffin embedded histology slides (Figs. 3a-b) we have also performed new experiments to monitor the possible off-target effects by flow cytometry (Fig. 3c).

1 and 5 μg VEGFC mRNA-LNP was administrated to the ear (or in a separate experiment to the back skin) and lymphatic growth was monitored in the ears, lung and small intestine of the same animal by fluorescent microscopy and flow cytometry 21 days after the back skin or ear injection. Of note, we have also tested higher dose of VEGFC mRNA-LNP (5 μg) in the revised manuscript. Taken together, our results indicate that VEGFC mRNA-LNP treatment induces lymphatic growth specifically in the injected organ, and not in other organs including the other (non-injected) ear, lung and small intestine (Fig. 3).

Reviewer's comment: 5. In Figure 4, the authors showed therapeutic effects of VEGFC mRNA-LNPs using a genetic secondary lymphedema model (FLT4-CreERT2; iDTRfl/fl mice). The experimental model is innovative, but data were not sufficient to support the author's conclusion on the therapeutic effects of VEGFC mRNA LNPs in the lymphedema model. Detailed experimental design also should be provided in the manuscript based on the paper published by Gardenier et al., in 2016, JCI Insight. (e.g. measurement of foot diameter, fibroadipose area, lymphatic capillary density, and lymphatic function using NIR lymphoscintigraphy). Furthermore, the authors should provide supplementary data to address whether the experimental lymphedema model is established well.

We agree with the Reviewer that these are very important points and performed a large number of experiments to address these issues in a highly rigorous and definitive fashion. We have optimized

the experimental setup that we used for inducing lymphedema in the Flt4-CreER^{T2}; *iDTR^{fl/fl}* genetic secondary lymphedema model. After optimizing the model, we increased the dose of diphtheria toxin to 60 ng per injection applied 3 times daily into the hind paw in these studies (30 ng diphtheria toxin was used in the original version of the manuscript). We also decided not to include the studies on the ear because the hind limb is a much better model for studying the development of secondary lymphedema. In the revised manuscript we show the diameter and clinical score changes of the hind paw after PBS and diphtheria toxin injections (Fig. 6a). Routine staining of histology slides also confirmed the development of secondary lymphedema in the model (Fig. 6b). Our data also reveal the increase of the fibroadipose area and Collagen I deposition in diphtheria toxin-treated Flt4-CreER^{T2}; *iDTR^{fl/fl}* mice and demonstrate the efficient elimination of lymphatic endothelial cells in the model (Fig. 6c-e). Importantly, the drainage of labelled macromolecules was blocked to the popliteal lymph node in Flt4-CreER^{T2}; *iDTR^{fl/fl}* mice 30 and 75 days after the diphtheria toxin treatment (Fig. 6f). In conclusion, we claim that we have established and rigorously validated a genetic secondary lymphedema mouse model.

Reviewer's comment: 6) In addition, the time point of measuring lymphedema is quite wrong. Lymphedema is usually a chronic process so the end point should be at least several weeks or months later to compare the secondary changes in dermis and subcutaneous tissues.

We appreciate the Reviewer's comment. We have addressed this question extensively in our revised manuscript. First, we have optimized the disease induction in our genetic secondary lymphedema model as it is described above. To this end, higher dose of diphtheria toxin was injected in the experimental animals. As it is shown in Fig. 7, VEGFC mRNA-LNP treatment effectively reduced diameter of the paw, clinical score, fibroadipose area and collagen I content compared to the Poly(C) RNA-LNP control 30 and 75 days after the treatment (Figs. 7a-d). VEGFC mRNA-LNP also induced the growth of the lymphatic vessels (Fig. 7e). This newly formed lymphatic network, induced by the VEGFC mRNA-LNP treatment, appeared to be fully functional because the drainage of labelled macromolecules to the popliteal lymph node was enhanced at 30 and 75 days after the treatment compared to the Poly(C) RNA control injection (Fig. 7f). Of note, lymphatic structures can be detected after 75 day in the Diphtheria toxin treated group, but these vessels do not form a functional network as it was also described (Gardenier et al., JCI Insight, 2016, PMID: 27699240). We believe that these experiments significantly improve the revised manuscript indicating that the VEGFC mRNA-LNP platform effectively reverses secondary lymphedema and restores lymphatic function.

Reviewer's comment: 7. No western blot was performed to confirm the protein production of VEGFC-mRNA-LNP in vivo.

As suggested by the Reviewer, we have monitored the *in vivo* VEGFC protein production using Western blot analysis and, additionally, ELISA. In these experiments higher level of VEGFC protein was detected in the interstitial fluid of the ear injected with VEGFC mRNA-LNPs compared to the Poly(C) RNA-LNP control (Figs. 1b-c).

Reviewer's comment: 8. In Figure 4i, in the Diphtheria toxin model, LYVE1 is still expressed, which is different from the description of the authors. Given the difference of DAPI expression between PBS and Diph toxin groups, the low staining of LYVE1 is due to the difference in imaging

or staining quality. Thus it is not clear whether their genetic secondary lymphedema model is appropriate.

We completely agree with the Reviewer that the quality of the DAPI and immunostaining staining were variable in the original experiments when we showed the histology of the hind paw. In the original manuscript we used OSTEOMOLL solution (Merck) which contains HCl to facilitate the decalcification before paraffin-based histology. To strengthen this part of the manuscript, we have optimized the decalcification protocol, and we started using the EDTA-based OSTEOSOFT solution (Merck) for the decalcification of the hind limbs in the new experiments. In the revised manuscript we significantly improved the quality of the immunostaining and DAPI labelling of our decalcified histology samples. In addition, the morphology of lymphatic vessels is shown by double staining of lymphatic endothelial markers (LYVE1 and Podoplanin) (Figs. 6e and 7e).

Reviewer's comment: 9. There is no direct evidence in vivo that VEGFC-mRNA-LNP is really proliferating lymphatic vessels. This should be performed by double-staining with a lymphatic marker and a proliferation marker such as BrdU or Ki67.

We agree the Reviewer that it is important to demonstrate that the VEGFC mRNA-LNP platform induces the proliferation of lymphatic endothelial cells in vivo. To this end, 5 days post VEGFC mRNA-LNP treatment, experimental mice were injected with EdU to monitor the proliferation of lymphatic endothelial cells. In the revised manuscript representative images, the number of the EdU positive lymphatic endothelial cells and the calculated mitotic index are shown (Figs. 2b-c). These results indicate that VEGFC mRNA-LNP treatment indeed induces strong proliferation of lymphatic endothelial cells in the injected organ.

Reviewer's comment: 10. No quantification data were provided in lymphatic vessels data measured in histology. Most of the conclusions were made by the representative images.

As suggested by the Reviewer, we have quantified most of the parameters in more detail in the revised manuscript. Quantification has been included in the revised figures and supplementary figures.

Reviewer's comment: Other minor points include: 1. Please show VEGFC expression levels after transfection of VEGFC-mRNA LNPs into HEK293 cells at various time points in Figure 1a.

We thank the Reviewer for this useful comment. In the revised manuscript we show the expression of VEGFC and GFP at various time points (8 hours, 1, 4 and 8 days) after adding VEGFC mRNA-LNP and GFP mRNA-LNP to cultured HEK293 cells (Fig. 1a and Suppl. Fig. 1a). Importantly, VEGFC expression was detectable 1 day after the VEGFC mRNA-LNP transfection, and the protein is present in a high amount in the supernatant after 4 and 8 days.

Reviewer #3

Reviewer's comment: This manuscript explores the potential of nucleoside-modified RNA encoding VEGFC to promote growth of lymphatic vessels and reverse experimental lymphoedema in mice. Lymphangiogenic proteins, such as VEGF-C, have been used many times in the past for

attempts to achieve this, but with mixed results. The novelty here is not the pro-lymphangiogenic strategy but the use of nucleoside modified mRNA as the delivery vehicle. Overall the results are clearly documented and impressive. However the manuscript is relatively silent on the nature of the changes to lymphatic vessels. My specific questions are outlined below. 1. What is the nature of the change in lymphatics induced by the nucleoside-modified RNA encoding VEGFC? For example, Figure 1d-g shows graphs about “normalized length of the lymphatic network”. What does this mean? Does it mean the average length of lymphatics? This is a key to understanding how the RNA elicits its effects. Were the lymphatics of wider caliber? What type of lymphatic vessel was influenced by the RNA? Are there more branch points? How is the morphology altered? A more thorough morphological analysis is required.

We thank the Reviewer for the positive and very useful comments. The organ-specific effect of the VEGFC mRNA-LNP treatment on lymphatic growth has been characterized and discussed in more detail in the revised manuscript. The total length of the lymphatic network, the average diameter of the lymphatic vessels and the number of the branching points per vision field were determined for the ear and the back skin (Fig. 2 and Suppl. Figs 2-3). The results of the quantification for dose and time dependence are discussed in the revised manuscript. The VEGFC mRNA-LNP treatment induces the proliferation of lymphatic capillaries, and we have detected an increase in the number of the lymphatic vessels with smooth muscle coverage in the VEGFC mRNA-LNP treated samples (Fig. 5c) suggesting that VEGFC mRNA-LNP treatment may induce the proliferation of collecting lymphatics. These vessels also appeared to be EdU positive (Fig. 5d). Thereafter, we have analyzed the function of the newly formed lymphatic network. Our results indicate that lymphatic vessels, which growth was induced by the VEGFC mRNA-LNP platform, are fully functional in the ear and hind limb (Figs. 5a-b). Furthermore, the VEGFC mRNA-LNP platform induces the growth of functional lymphatic vessels and reduces secondary lymphedema by restoring lymphatic function (Fig. 7).

Reviewer’s comment: 2. In Figure 2 the Authors explore the systemic effects of their approach. But they use only 0.2 microgram which is a lower dose than in other parts of the study. And they examine the effects in lung and small intestine after only 4 hours. Why not use the 1 microgram dose and give more time for effects on lymphatics at distant sites to become apparent? This seems not to be a sufficiently rigorous assessment of systemic effects.

We completely agree with the Reviewer that these are important points and this part of the original manuscript was not clear as it was also pointed out by Reviewer #2. In the revised manuscript we have clarified that 1 μ g GFP mRNA-LNP was injected into the ear and GFP expression was monitored in the ear, lung and small intestine 4 hours after the injection (Suppl. Fig. 4). Our results indicate that GFP expression can be detected at the injection site, but not in other organs including the lung and the small intestine in the same animal. These results are in agreement with our previous report (Pardi et. al, Journal of Controlled Release, 2015, PMID: 26264835), in which the different delivery routes of the mRNA-LNP platform were compared in a luciferase reporter assay in vivo. We agree with the reviewer that using a higher dose of VEGFC mRNA-LNP is more appropriate to test the possible systemic effect of our novel platform. Thus, 1 or 5 μ g VEGFC mRNA-LNP was administrated to the ear (or in a separate experiment to the back skin) and lymphatic growth was monitored not only in whole-mount organs and histology slides by immunofluorescence, but by flow cytometry at 21 days in the ears, lung and small intestine of the

same animal. Taken together, our results indicate that VEGFC mRNA-LNP treatment induces lymphatic growth in the injected organ, and not in other organs including the other ear (non-injected), lung and small intestine (Fig. 3).

Reviewers' Comments:

Reviewer #1:

Remarks to the Author:

The authors addressed my concerns and comments by adding new experimental data and descriptions.

Reviewer #2:

Remarks to the Author:

This paper has technical merits and has translational potential while all the other parts were the reproduction of what has been shown by many papers.

Many experimental data were presented; however, overall, data images for the lymphatic vessels in vivo are unconvincing and insufficient to support their conclusion.

1. Images for Prox1-GFP in diaphragm of Poly(C) RNA-LNP treated mouse in Figure 2g are very ambiguous. Most of the immunohistochemistry data are not sufficient due to their strong backgrounds and low resolution (e.g. Figure 2g, podoplanin in Figure 3b and 4a, overall whole mount image data, Figure 5...etc). It is unable to draw accurate conclusion with the given data.
2. In Figure 1a and 1b, the authors showed multiple bands of VEGFC. Please explain which band (upper or lower, or both) represents VEGFC. Please indicate VEGFC exactly using any symbol and describe those in the Figure Legend section. Data in Figure 1a and 1b showed different sizes of VEGFC proteins in HEK293T cells and ears.
3. In Figure 1a, the authors should determine the protein expression later than 8 hours after treating VEGFC mRNA-LNPs into HEK293T cells. Please demonstrate how long it takes for VEGFC protein to be expressed by VEGFC mRNA-LNPs. It would be helpful to demonstrate the stability of VEGFC protein expressed by VEGFC mRNA-LNPs in vitro as well as in vivo.
4. Figure 2a showed VEGFC mRNA-LNPs induced lymphatic vessel growths. However, the vessels showed unstable morphology. The proliferation and normalization of lymphatic vessels are both important for lymphatic system regeneration. If VEGFC mRNA-LNPs could induce lymphangiogenesis and lymphatic vessel normalization in long-term follow up, please show any long-term data in vivo.
5. In Figure 2b, the EdU positive cells colocalized with DAPI and LYVE signals are not convincing. They should demonstrate individual panels of immunostained figures to clearly demonstrate them with confocal microscopy (z-stack reconstruction).
6. There are only quantification graphs in Figure 2e and 2f. Please provide representative images in the Figure.
7. In Figure 4b, the authors mentioned that VEGFC mRNA-LNPs have no effects in blood vessel proliferation by ear whole mount staining for CD31-positive blood vessels. However, CD31 is unsuitable to use as a single marker to distinguish between blood vessels and lymphatic vessels, because CD31 is also expressed in lymphatic vessels with varying densities. Other blood vessel-specific markers should be used or added to better demonstrate the separate density of lymphatic and blood vessels.
8. Figure 4, it is now well known that VEGFC is also angiogenic for blood vessels: proliferating blood vessels. Their data conflict with those results. How would they reconcile these? Have they checked VEGFR2 activation in the cells and tissues?
9. The authors conducted experiment to monitor lymphatic functions in vivo using 70kDa tetramethylrhodamine-dextran (rh-D). In Figures 6 and 7, intradermal paw injection data did not demonstrate quantitative results. Please quantify intensity of injected tetramethylrhodamine-dextran in injection sites and Rh-D positive popliteal lymph nodes. The data in the manuscript appeared that strong signals of Rh-D were transferred into popliteal lymph nodes.
10. In Figure 6 and 7, the authors should provide quantification data on histological analysis of the lymphatic vessels. Most of the conclusions made by the authors were based on the representative images

Reviewer #3:

Remarks to the Author:

The Authors have satisfactorily addressed the questions and criticisms I raised about the original version of the manuscript.

Ms. Ref. No.: NCOMMS-19-01080B (Revision #2)

Title: Nucleoside-Modified VEGFC mRNA Induces Organ-Specific Lymphatic Growth and Reverses Experimental Lymphedema

We would like to thank the Reviewers for the thoughtful analysis of our study and helpful comments. We appreciate that Reviewers #1 and #3 think that we addressed their concerns and comments by adding new experimental data and descriptions in the first revision of our manuscript. To address the specific points, we included the results of a large number of additional experiments and statistical calculations in our second revision. We also added 11 new figure panels and replaced 4 panels with new representative images. Even though the current SARS-CoV-2 pandemic affects us significantly (the first author had to spend weeks in quarantine during this revision process) we believe that we have addressed all the points of Reviewer #2 in a highly rigorous and definitive fashion. Importantly, the inclusion of all the additional experiments and data suggested has significantly improved the revised manuscript. The major revisions to the manuscript that address these points include the following:

1. We rigorously characterized the production and release of VEGFC *in vitro* and *in vivo* (Fig. 1 and Suppl. Fig. 2). Importantly, VEGFC level peaks 1 day after VEGFC mRNA-LNP treatment *in vivo* and the protein is detectable for an extended period of time (15 days) in the ear interstitial compartment (Fig. 1a).
2. We showed that LYVE-1 and EdU signal overlap each other 5 days after VEGFC mRNA-LNP treatment using confocal images and Z-stack confocal imaging suggesting the proliferation of lymphatic endothelial cells (Fig. 3b).
3. We quantified the number and function of lymphatic vessels in our secondary lymphedema model 30 and 75 days after the first Diphtheria toxin treatment (Figs. 8e, 9b) to demonstrate that our lymphedema model has been rigorously validated.
4. We quantified the number and function of lymphatic vessels after Poly(C) and VEGFC mRNA-LNP treatment 30 and 75 days after the first Diphtheria toxin treatment in our secondary lymphedema model (Figs. 10e, 11b). Our results indicate that nucleoside-modified VEGFC mRNA-LNP treatment reverses experimental lymphedema *in vivo*.

Importantly, all the new experiments confirmed our original conclusions that 1) describe a novel *in vivo* application of the nucleoside-modified mRNA-LNP therapeutic protein delivery platform, 2) describe an approach for identifying the organ-specific physiological and pathophysiological roles of the lymphatic system, and importantly 3) propose an efficient and safe treatment option that may serve as the basis for the development of a novel therapeutic tool to cure or reduce lymphedema. Our study describes the first use of the nucleoside-modified mRNA in lymphatic diseases, and based on the data presented in the manuscript we believe that the use of this platform is a viable strategy for the treatment of secondary lymphedema.

We again thank the reviewers for their thoughtful comments and questions. Our responses to the specific points are provided below. Corresponding changes in the manuscript are highlighted by

blue font color (the changes for the first revision are highlighted in red font color). We hope that our study is now acceptable for publication in Nature Communications.

Reviewers' comments:

Reviewer #1

Reviewer's comment: The authors addressed my concerns and comments by adding new experimental data and descriptions.

We appreciate the feedback from Reviewer #1 and thank the helpful comments and suggestions.

Reviewer #3

Reviewer's comment: The Authors have satisfactorily addressed the questions and criticisms I raised about the original version of the manuscript.

We would like to thank the Reviewer #3 for the useful criticisms and questions.

Reviewer #2

Reviewer's comment: This paper has technical merits and has translational potential while all the other parts were the reproduction of what has been shown by many papers. Many experimental data were presented; however, overall, data images for the lymphatic vessels in vivo are unconvincing and insufficient to support their conclusion.

We would like to thank the Reviewer for the kind comment about the technical merits of our work, and we are very glad that he/she thinks that our manuscript holds significant translational impact. In this project our goal was to develop a novel, state-of-the-art platform for identifying the organ-specific physiological and pathophysiological roles of the lymphatic system and propose an efficient and safe treatment option that may serve as the basis for the development of a novel therapeutic tool to cure or reduce lymphedema. We strongly believe that our highly efficient platform will be an ideal tool to reveal not only the novel organ-specific functions of the VEGFC-VEGFR3 axis in the lymphatic system but other unexpected roles of the lymphangiogenic factor (e.g. in spiral artery remodeling, Pawlak et al., J. Clin. Invest., 2019; PMID: 31415243) in future studies.

To address the questions and comments of the Reviewer we added 11 new figure panels (Figs. 1a, 2b, 3b, 6c, 7e, 8e, 9b, 10e, 11b and Suppl. Figs. 2a-c), replaced 4 panels with new representative images (Fig. 2a, Fig. 4a, Fig. 5b, Fig. 6a), and performed a large number of new experiments. We indeed feel that the additional experiments helped to expand the project and has significantly improved the manuscript and the image quality of the figures.

Importantly, we strongly believe that there are a number of critically important novel aspects of the experiments described within this manuscript: Our current studies 1) describe a novel *in vivo* application of the nucleoside-modified mRNA-LNP therapeutic protein delivery platform, 2) describe an approach for identifying the organ-specific physiological and pathophysiological roles of the lymphatic system, and importantly 3) propose an efficient and safe

treatment option that may serve as the basis for the development of a novel therapeutic tool to cure or reduce lymphedema. We tried to emphasize the above-mentioned novel aspects of our work in the revised manuscript.

Reviewer's comment: 1. Images for Prox1-GFP in diaphragm of Poly(C) RNA-LNP treated mouse in Figure 2g are very ambiguous. Most of the immunohistochemistry data are not sufficient due to their strong backgrounds and low resolution (e.g. Figure 2g, podoplanin in Figure 3b and 4a, overall whole mount image data, Figure 5...etc). It is unable to draw accurate conclusion with the given data.

We appreciate the Reviewer's comment. We replaced and added new images (previously Fig. 2b) to show the proliferation of lymphatic vessels in the diaphragm after intraperitoneal VEGFC mRNA-LNP treatment (Fig. 4). Anti-LYVE1 and anti-Podoplanin immunostaining protocols were used to visualize the lymphatic vessels in the organ. We also replaced and added new representative images to demonstrate the lymphatic growth in the lung after intratracheal and in the musculus gastrocnemius after intramuscular treatment with VEGFC mRNA-LNP (Fig. 4). We believe that the new data set clearly supports our conclusions.

We agree with the Reviewer, therefore, we also replaced the representative images in old Figs. 3b and 4a (new Figs. 5b and 6a). We also replaced or included new whole-mount images to visualize the lymphatic and blood vessels after VEGFC mRNA-LNP treatment (Figs. 2a, 2b and 6c). Please note that Prox1-GFP labels lymphatics, CD31 is a panendothelial marker expressed both on blood and lymphatic vessels and vWF staining is shown as a blood vessel marker.

In total, we added 11 new figure panels (Figs. 1a, 2b, 3b, 6c, 7e, 8e, 9b, 10e, 11b and Suppl. Figs. 2a-c) and replaced 4 panels with new representative images (Fig. 2a, Fig. 4a, Fig. 5b, Fig. 6a) to improve the quality of our representative images.

Reviewer's comment: 2. In Figure 1a and 1b, the authors showed multiple bands of VEGFC. Please explain which band (upper or lower, or both) represents VEGFC. Please indicate VEGFC exactly using any symbol and describe those in the Figure Legend section. Data in Figure 1a and 1b showed different sizes of VEGFC proteins in HEK293T cells and ears.

Thank you for the comment. The secreted VEGFC is proteolytically processed, which process influences the molecular weight. It has been reported by others the processed VEGFC is present in the 25-35 kDa molecular weight range showing multiple bands (Bui et al., J Clin Invest, 2016; PMID: 27159393; Jeltsh et al, Circulation, 2014, 24552833). This process is mediated by CCBE1 and ADAMTS3. The processing of VEGFC is different in different cell lines *in vitro* and it is tissue specific *in vivo*. Please note that it is challenging to detect the VEGFC protein *in vivo and vitro*, especially using Western blot, most labs use tagged VEGFC to monitor the expression and processing (Bui et al., J Clin Invest, 2016; PMID: 27159393).

To monitor VEGFC expression in an independent assay we performed anti-VEGFC ELISA measurements. As it is shown in Suppl. Figs. 2b-c VEGFC was detectable in HEK293 cell lysates after VEGFC mRNA-LNP treatment showing a peak at 1 day, and after that the protein was present in the cell culture supernatant for an extended period *in vitro*. Critically, we also detected the presence of VEGFC in the interstitial fluid of ear samples after VEGFC mRNA-LNP injection at 1, 5, 10 and 15 days (Fig. 1a). VEGFC protein expression peaked 1 day and the VEGFC expression was significantly higher compared to the control as long as 15 days after VEGFC mRNA-LNP

injection. Overall, we provided valuable insights about the kinetics of VEGFC protein expression from nucleoside-modified mRNA-LNPs *in vitro* and *in vivo*.

Reviewer's comment: 3. In Figure 1a, the authors should determine the protein expression later than 8 hours after treating VEGFC mRNA-LNPs into HEK293T cells. Please demonstrate how long it takes for VEGFC protein to be expressed by VEGFC mRNA-LNPs. It would be helpful to demonstrate the stability of VEGFC protein expressed by VEGFC mRNA-LNPs in vitro as well as in vivo.

We appreciate the Reviewer's excellent comment. As mentioned above, we assessed VEGFC protein expression at various time points *in vitro* and *in vivo*. We believe that the data generated is clean and significantly improves our manuscript. As it is shown in Suppl. Figs. 2a-c, VEGFC was detectable in HEK293 cell lysates after VEGFC mRNA-LNP treatment showing a peak at 1 day, and after that the protein was present in the cell culture supernatant for an extended period of time. We also detected the presence of VEGFC in the interstitial fluid of ear samples after VEGFC mRNA-LNP injection at 1, 5, 10 and 15 days (Fig. 1a). VEGFC protein expression peaked 1 day and the VEGFC expression was significantly higher compared to the control as long as 15 days after VEGFC mRNA-LNP injection.

Reviewer's comment: 4. Figure 2a showed VEGFC mRNA-LNPs induced lymphatic vessel growths. However, the vessels showed unstable morphology. The proliferation and normalization of lymphatic vessels are both important for lymphatic system regeneration. If VEGFC mRNA-LNPs could induce lymphangiogenesis and lymphatic vessel normalization in long-term follow up, please show any long-term data in vivo.

We agree with the Reviewer, it is a conceptual question whether VEGFC mRNA-LNP treatment results in a long-term and maintained proliferation of the lymphatic vessels. To this end, we show the morphology of lymphatic vessels not only after 22, but also 60 days after the VEGFC mRNA-LNP injection compared to the control in the revised manuscript (Fig. 2b). The lymphatic growth was also quantified by measuring the length of the lymphatic network, the average diameter of the lymphatic vessels, and counting the number of the branching points after VEGFC mRNA-LNP injection. We measured significant increases in all three parameters, and, importantly, the effect of VEGFC mRNA-LNP treatment was maintained for an extended period of time (up to 60 days) both in the ear and back skin (Fig. 2c and Suppl. Figs. 3a-b).

In addition, we demonstrated that lymphatic vessels take up and transport large molecular weight molecules in both the ears and hind limbs after control treatment (Figs. 7a-b), indicating this method is a good tool to monitor lymphatic function in the ear and hind limb. Importantly, increased number of lymphatic vessels can be detected in the VEGFC mRNA-LNP injected organs, and this dense network of lymphatic vessels also takes up and transports the 70 kDa Rh-D without signs of significant leakage of the macromolecule 22 days in the ear and 75 days after the treatment (Figs. 7a-b). These results indicate that VEGFC mRNA-LNP treatment not only induce the proliferation of lymph vessels but these new vessels are also functional even after an extended period of time.

Importantly, a single low dose of VEGFC mRNA-LNPs effectively reduced paw thickness, clinical score, the fibroadipose area and Collagen I content compared to the control 30 and 75 days after disease induction in the experimental secondary lymphedema mouse model (Figs. 10a-c, f).

VEGFC mRNA-LNP stimulated the formation of a functional network of lymphatic vessels (Figs. 10d-e), which was efficient to drain macromolecules from the paw to the popliteal lymph nodes in the VEGFC mRNA-LNP treated animals 30 and 75 days after lymphedema induction (Figs. 11a-b).

Overall, we believe that we provide compelling data for the reviewer's request and these important results demonstrate that administration of a single low dose of VEGFC mRNA-LNPs is an efficient strategy to induce the formation of functional lymphatic vessels, which then reverse the development and progression of secondary lymphedema (Figs. 10-11).

Reviewer's comment: 5. In Figure 2b, the EdU positive cells colocalized with DAPI and LYVE signals are not convincing. They should demonstrate individual panels of immunostained figures to clearly demonstrate them with confocal microscopy (z-stack reconstruction).

We would like to thank the Reviewer for the comment. In the revised manuscript proliferation of the lymphatic endothelial cells by determining the number of 5-ethynyl-2'-deoxyuridine (EdU) positivity is also shown by confocal imaging (Fig. 3b). Both the number and fraction of EdU positive nuclei of lymphatic endothelial cells were highly increased in ears injected with VEGFC mRNA-LNP compared to the control (Figs. 3a-c, Suppl. Videos 1-2). We also show the proliferation of collecting lymphatic vessels using EdU incorporation by confocal imaging (Fig. 7e).

Reviewer's comment: 6. There are only quantification graphs in Figure 2e and 2f. Please provide representative images in the Figure.

Thank you for the comment. We included additional representative images in the revised manuscript to demonstrate the effect of VEGFC mRNA-LNP injection compared to the Poly(C) RNA-LNP control 60 days after the treatment (Fig. 2b).

Reviewer's comment: 7. In Figure 4b, the authors mentioned that VEGFC mRNA-LNPs have no effects in blood vessel proliferation by ear whole mount staining for CD31-positive blood vessels. However, CD31 is unsuitable to use as a single marker to distinguish between blood vessels and lymphatic vessels, because CD31 is also expressed in lymphatic vessels with varying densities. Other blood vessel-specific markers should be used or added to better demonstrate the separate density of lymphatic and blood vessels.

We completely agree with the reviewer and, thus, we assessed the effect of VEGFC mRNA-LNP treatment on blood vessel formation by performing immunostaining with the pan-endothelial marker CD31 and blood vessel-specific marker von Willebrand Factor (vWF) of paraffin-based histology slides and whole-mount samples (Figs. 6a-c and Suppl. Fig. 6). We also quantified the number of the vWF positive blood vessels after Poly(C) RNA-LNP and VEGFC mRNA-LNP treatment. Our new data set further confirmed that VEGFC mRNA-LNPs did not result in local blood vessel proliferation, while the VEGFC mRNA-LNP-induced lymphatic growth was present shown by LYVE1 and Podoplanin immunostaining of lymphatic endothelial cells (Figs. 6a-c and Suppl. Fig. 6).

Reviewer's comment: 8. Figure 4, it is now well known that VEGFC is also angiogenic for blood vessels: proliferating blood vessels. Their data conflict with those results. How would they reconcile these? Have they checked VEGFR2 activation in the cells and tissues?

We agree with the Reviewer that it has been reported that VEGFC induces some proliferation and dilation of blood vessels and a slight increase in their permeability but these findings were not confirmed by others (PMID references: 21282502, 26018927, 12235206, 12087065, 9826710). It is possible that some of these effects were not induced by the VEGFC directly, but they were elicited by the adenoviral delivery of the lymphangiogenic factor, or require special conditions. It should be noted that we also detected a transient and slight increase in blood vessel permeability during the early phase after VEGFC mRNA-LNP treatment (data not shown) suggesting a possible transient effect of VEGFC on blood vessel permeability.

Importantly, we found that VEGFC mRNA-LNP treatment resulted in no increased blood vessel proliferation shown by vWF immunostaining (Fig. 6). This is in contrast to the findings with adenovirus-based delivery of genes (PMID references: 28592330). Our findings are in accordance with previous studies which demonstrated that VEGFC, in contrast to VEGFD which induces both blood and lymph vessel growth in adults, did not have a major impact on blood vessel proliferation (Figs. 6a-c) (PMID references: 12235206, 12714562, 17362136, 12087065). It should be noted that nucleoside-modified VEGFC mRNA-LNP treatment may induce the stimulation of VEGFR2, but this process does not lead to significant proliferation of blood vessels. Further studies will need to be performed to understand the effect of our platform on VEGFR2 signaling.

Reviewer's comment: 9. The authors conducted experiment to monitor lymphatic functions in vivo using 70kDa tetramethylrhodamine-dextran (rh-D). In Figures 6 and 7, intradermal paw injection data did not demonstrate quantitative results. Please quantify intensity of injected tetramethylrhodamine-dextran in injection sites and Rh-D positive popliteal lymph nodes. The data in the manuscript appeared that strong signals of Rh-D were transferred into popliteal lymph nodes.

We agree with the Reviewer and, thus, we quantified the lymphatic function in our secondary lymphedema model (old Figs. 6 and 7). In addition, we performed more independent experiments to validate the genetic model and test the *in vivo* effect of VEGFC mRNA-LNP treatment in the lymphedema model.

We injected 70 kDa Rh-D into the hind paw in equal amount in each group, and monitored the lymphatic function 90 minutes after the tracer injection. We quantified the transport of the fluorescently labelled macromolecule to the popliteal lymph nodes, which parameter represents the function of lymphatics in our secondary lymphedema model. Importantly, the drainage of labelled macromolecules to the popliteal lymph node was also blocked in our genetic lymphedema model 30 and 75 days after the Diphtheria toxin treatment (Figs. 9a-b). We demonstrated that we have set up a genetic secondary lymphedema model.

This newly formed lymphatic network – induced by the VEGFC mRNA-LNP treatment – appeared to be fully functional because drainage of labelled macromolecules to the popliteal lymph node was enhanced at 30 and 75 days after the treatment compared to the Poly(C) RNA-LNP control injection indicated by representative images and the quantification of the transport to the

popliteal lymph nodes (Figs. 11a-b). Taken together, our results clearly indicate that nucleoside-modified VEGFC mRNA-LNP treatment reverses experimental lymphedema *in vivo*.

Reviewer's comment: 10. In Figure 6 and 7, the authors should provide quantification data on histological analysis of the lymphatic vessels. Most of the conclusions made by the authors were based on the representative images

As suggested by the Reviewer, we quantified not only the function but also the number of lymphatic vessels in our *in vivo* models. We demonstrated that the lymphatic vessels are efficiently eliminated in our secondary lymphedema model (Fig. 8e). In addition, VEGFC mRNA-LNP induced the growth of the lymphatic vessels in the secondary lymphedema model (Fig. 10e). Our results indicate that nucleoside-modified VEGFC mRNA-LNP treatment reverses experimental lymphedema *in vivo*.

We appreciate the comments of Reviewer #2 and we strongly believe that our manuscript benefitted from his/her multiple suggestions. We have addressed each of the concerns he/she raised and we hope that our manuscript is now suitable for publication in Nature Communications.

Reviewers' Comments:

Reviewer #2:

Remarks to the Author:

In the manuscript, the authors claimed that the nucleoside-modified mRNA-LNP is a novel therapeutic platform to deliver protein demonstrating the potential therapeutic effects of VEGFC mRNA-LNPs in experimental lymphedema through VEGFC-induced lymphangiogenesis. Although the approach is technically innovative and the study has a translational potential, much data are descriptive and are not convincing to support their conclusion.

1. In the Introduction section, advantages of therapeutic approaches using modified mRNA-based platform and encapsulated mRNA in lipid nanoparticles (LNP) were described. However, the justification for using LNP was insufficient. (e.g. Why the authors used LNP for encapsulation of VEGFC-mRNA, not just VEGFC-mRNA, What is the advantages and characteristics of LNP.. etc.)
2. Also, many experiments miss the comparison in vivo and in vitro data between VEGFC-mRNA alone and VEGFC-mRNA-encapsulated in LNP, which would be the key for claiming the advantages of their technology. It is because VEGF-C is well known for its lymphangiogenic activities and its therapeutic effects on lymphedema.
3. Is the LNP biodegradable in vivo? If that is the case, please show the results using experimental data.
4. The authors quantified lymphatic vessel numbers per view field of 2-3 images taken per a histological slide in the manuscript. However there is no description on how many animals were used for quantitative analysis. Please add detailed description of the quantification methods in the Method section or the Figure legend.
5. In Figure 2d, representative images are missing. Please add those used for the quantification graphs in Figure 2.
6. In Figure 4, the authors should add a quantification graph of lymphatic growth by injection of VEGFC mRNA-LNP in vivo. The quantitative data can strongly support the conclusion.
7. It would be better to combine Figures 2, 3, and 4. They are demonstrating the same conclusion that VEGFC mRNA-LNP augment lymphangiogenesis in vivo.
8. In the upper images of Figure 6c, is CD31 indicating only blood vessels or both blood and lymphatic vessels? In either case, there is no correlation between representative images and quantification graphs.
9. The authors conducted experiments to monitor lymphatic functions in vivo using 70kDa tetramethylrhodamine-dextran (Rh-D). In Figure 11, the imaging data are unclear to support that the popliteal signals are stronger in VEGFC mRNA-LNP treated ones. Also, describe how the quantitative data were calculated in the Method section or Figure legend.
10. There is no direct evidence showing proliferative lymphatic vessels. They should include double staining data showing proliferative markers and lymphatic vessels marker(s).

REVIEWER COMMENTS

Reviewer #2 (Remarks to the Author):

In the manuscript, the authors claimed that the nucleoside-modified mRNA-LNP is a novel therapeutic platform to deliver protein demonstrating the potential therapeutic effects of VEGFC mRNA-LNPs in experimental lymphedema through VEGFC-induced lymphangiogenesis. Although the approach is technically innovative and the study has a translational potential, much data are descriptive and are not convincing to support their conclusion.

We are thankful for Reviewer #2 for the commitment to help us to further improve the manuscript. To address the specific points, we included the results of additional quantifications and replaced figure panels in the third revision of our manuscript. We also answered the questions of Reviewer #2 below. In summary, we added 2 new figure panels (Figure 2f and Suppl. Figure 3d), replaced 1 figure panel (Figure 10a) with new representative images and combined Figure 2 and Figure 4 as recommended. Even though the current SARS-CoV-2 pandemic still affects us significantly, we believe that we have rigorously addressed all the points of Reviewer #2.

Importantly, all the new experiments confirmed our original conclusions that 1) describe a novel *in vivo* application of the nucleoside-modified mRNA-LNP therapeutic protein delivery platform, 2) describe an approach for identifying the organ-specific physiological and pathophysiological roles of the lymphatic system, and importantly 3) propose an efficient and safe treatment option that may serve as the basis for the development of a novel therapeutic tool to cure or reduce lymphedema. Our study describes the first use of the nucleoside-modified mRNA-LNP system in lymphatic diseases, and based on the data presented in the manuscript, we believe that the use of this platform is a viable strategy for the treatment of secondary lymphedema.

We again thank Reviewer #2 for his/her thoughtful comments and suggestions. Our responses to the specific points are provided below. Corresponding changes in the manuscript are highlighted by green font color (the changes are highlighted in red font color for the first revision, in blue font color for the second revision). We hope that our study is now acceptable for publication in Nature Communications.

1. In the Introduction section, advantages of therapeutic approaches using modified mRNA-based platform and encapsulated mRNA in lipid nanoparticles (LNP) were described. However, the justification for using LNP was insufficient. (e.g. Why the authors used LNP for encapsulation of VEGFC-mRNA, not just VEGFC-mRNA, What is the advantages and characteristics of LNP.. etc.)

The LNP systems we use for delivery of mRNA represent a mature technology including a highly efficient encapsulation process, robust, scalable production and LNP stability for at least 1 year¹. The LNPs we use differ from the commercially available liposomes. Our LNPs have 4 components (ionizable cationic lipid, PEG lipid, cholesterol and phosphatidylcholine) that self-assemble to 80-100 nm particles after a rapid and controllable mixing process². A key safety feature of these LNPs is a relatively neutral surface exterior to avoid extensive binding to serum proteins after *in vivo* delivery. Incorporation of the ionizable cationic lipid into LNPs is critical to achieve very low surface charge under neutral pH (for example in the blood stream) but positive charge at low pH

of late endosomes. As commercial transfection reagents do not contain this component, most of them are not well-tolerated and ineffective *in vivo*. Limited amount of information is available about the specific cellular uptake mechanisms of LNPs but, based on early findings, RNA-loaded LNPs are taken up by endocytosis, go through the endosomal pathway, get disrupted by endosomal acidification, and a small portion of the RNA escapes from the endosomes to enter the cytosol, where protein production from mRNA occurs¹. Norbert Pardi has been collaborating with Acuitas Therapeutics –one of the pioneers of the LNP technology– for over 6 years. Acuitas provided LNP formulations for this study after performing rigorous LNP quality control (size and integrity) to avoid batch-to-batch variations.

The use of LNPs is critical because they facilitate uptake of mRNA and protect it from extracellular RNases. Pardi et al. demonstrated that intradermally administered LNP-formulated mRNA gets translated for a longer time than unformulated (naked) mRNA and more protein is produced from the same amount of LNP-formulated mRNA than from naked mRNA (Figure 1 of Pardi et al, J Exp Med, 2018)³. These are all critical features for therapeutic activity and justify the use of LNPs.

2. Also, many experiments miss the comparison in vivo and in vitro data between VEGFC-mRNA alone and VEGFC-mRNA-encapsulated in LNP, which would be the key for claiming the advantages of their technology. It is because VEGF-C is well known for its lymphangiogenic activities and its therapeutic effects on lymphedema.

Please see our answer for Q1. LNPs are critical for durable and high level protein production from mRNA. Pardi et al demonstrated that intradermally administered LNP-formulated mRNA gets translated for a longer period of time than the unformulated (naked) mRNA and a higher amount of protein is produced from the same amount of LNP-formulated mRNA than from naked mRNA (Figure 1 of Pardi et al, J Exp Med, 2018)³. The technology we use is well-established and safe (currently being evaluated in Phase III clinical trials), thus we do not believe that additional studies would need to be added to this manuscript.

In addition, we agree that VEGFC is a well-known lymphangiogenic factor. Our important finding is that nucleoside-modified RNA-based system encoding VEGFC is an efficient approach to induce lymphatic growth and improve lymphatic function *in vivo*.

3. Is the LNP biodegradable in vivo? If that is the case, please show the results using experimental data.

The nucleoside-modified mRNA-LNP technology is well-established and safe and multiple published studies demonstrated this⁴⁻¹¹. Currently, this leading technology is being used in clinical studies to develop SARS-CoV-2 vaccines^{12, 13, 14}. Most importantly, the first LNP-containing drug (patisiran) has recently been approved for human use by the FDA¹⁵. Thus, we do not believe that additional experiments demonstrating the safety of LNPs need to be added to this manuscript. Such studies are clearly out of the scope of this manuscript.

4. The authors quantified lymphatic vessel numbers per view field of 2-3 images taken per a histological slide in the manuscript. However, there is no description on how many animals were used for quantitative analysis. Please add detailed description of the quantification methods in the Method section or the Figure legend.

We apologize if the way how we presented our data was not clear for the reviewer in some cases. Now, we always indicate the number of the animals and independent experiments used in various experiments (information is added to the figure legends). We carefully reviewed the manuscript and added additional information to the revision, if it was necessary. For example the current form of Figure 2a is the following: “Representative images 22 days (a) and 60 days (b) after the ear treatment of 15 (a) and 5 (b) mice per group are shown by whole-mount fluorescent stereo microscopy.” In addition, we show bar charts with individual data points in our figure panels. We also included additional details about the quantifications in the methods section.

5. In Figure 2d, representative images are missing. Please add those used for the quantification graphs in Figure 2.

We added representative images to Suppl. Figure 3d as requested by Reviewer #2. The effect of 3 different concentrations of Poly(C) and VEGFC mRNA-LNPs was characterized in 6-17 animals in each group as it is shown in the quantification of the data (Figure 2d). The quantification of Figure 2d is based on 72 separate images from 6-17 mice per group in which the length of the lymphatic network, the average diameter of lymphatic vessels and the branching points per field of view were measured as described in the methods section.

6. In Figure 4, the authors should add a quantification graph of lymphatic growth by injection of VEGFC mRNA-LNP in vivo. The quantitative data can strongly support the conclusion.

We thank Reviewer #2 for the comment, we quantified the data and included the bar graphs in Figure 2 (Figure 2f). Importantly, the effect of the VEGFC mRNA-LNP treatment was significant in all organs compared to the Poly(C) RNA-LNP controls.

7. It would be better to combine Figures 2, 3, and 4. They are demonstrating the same conclusion that VEGFC mRNA-LNP augment lymphangiogenesis in vivo.

As requested by Reviewer #2 we combined the original Figures 2 and 4, and moved a part of the data to the Supplementary Figures (Suppl. Figure 3d). The current Figure 2 demonstrates the local effect of the VEGFC mRNA-LNP treatment on lymphatic proliferation *in vivo* in different organs. We would prefer to keep Figure 3 separate, in which we show that lymphatic endothelial cells are proliferating using an EdU assay combined with the immunostaining of lymphatic endothelial cells.

8. In the upper images of Figure 6c, is CD31 indicating only blood vessels or both blood and lymphatic vessels? In either case, there is no correlation between representative images and quantification graphs.

The effect of the VEGFC-mRNA-LNP treatment on blood vessel proliferation is based on the quantification of the vWF positive vessels in paraffin based histology sections of the ears (Figures 5a and b). We apologize if it was confusing how we performed these measurements. We clarified this in the current form of the manuscript. We also show anti-CD31 staining on the same tissues which labels both the blood and lymphatic endothelial cells (Figure 5a). In addition, we generated new whole-mount fluorescent and confocal images of the ear using lymphatic and blood vessel

markers, and moved them to Suppl. Fig. 7, where we show the effect of VEGFC-mRNA-LNP injection on lymphatic growth and proliferation of blood vessels compared to the poly(C) RNA-LNP control treatment.

9. *The authors conducted experiments to monitor lymphatic functions in vivo using 70kDa tetramethylrhodamine-dextran (Rh-D). In Figure 11, the imaging data are unclear to support that the popliteal signals are stronger in VEGFC mRNA-LNP treated ones. Also, describe how the quantitative data were calculated in the Method section or Figure legend.*

We thank Reviewer #2 the comment, we replaced the panel of Figure 10a (original Figure 11a) with another representative image. We also labeled the magnified area of popliteal lymph nodes in the revised manuscript (Figures 8 and 10). We apologize that the description of the quantification was not clear and hard to find. We also revised that part of the Methods section in the following paragraph “Quantification of microscopic images“.

10. *There is no direct evidence showing proliferative lymphatic vessels. They should include double staining data showing proliferative markers and lymphatic vessels marker(s).*

We demonstrated that VEGFC mRNA-LNP treatment induces local proliferation of lymphatic endothelial cells shown by immunostaining of lymphatic endothelial cell marker and using an EdU proliferation assay in Figure 3 shown by fluorescent and confocal imaging. The quantification of the data is also included in the manuscript (Figure 3c).

References

1. Cullis, P.R. & Hope, M.J. Lipid Nanoparticle Systems for Enabling Gene Therapies. *Mol Ther* **25**, 1467-1475 (2017).
2. Belliveau, N.M. et al. Microfluidic Synthesis of Highly Potent Limit-size Lipid Nanoparticles for In Vivo Delivery of siRNA. *Mol Ther Nucleic Acids* **1**, e37 (2012).
3. Pardi, N. et al. Nucleoside-modified mRNA vaccines induce potent T follicular helper and germinal center B cell responses. *J Exp Med* **215**, 1571-1588 (2018).
4. Pardi, N. et al. Zika virus protection by a single low-dose nucleoside-modified mRNA vaccination. *Nature* **543**, 248-251 (2017).
5. Pardi, N., Hogan, M.J., Porter, F.W. & Weissman, D. mRNA vaccines - a new era in vaccinology. *Nat Rev Drug Discov* **17**, 261-279 (2018).
6. Pardi, N. et al. Nucleoside-modified mRNA immunization elicits influenza virus hemagglutinin stalk-specific antibodies. *Nat Commun* **9**, 3361 (2018).
7. Pardi, N. & Weissman, D. Nucleoside Modified mRNA Vaccines for Infectious Diseases. *Methods Mol Biol* **1499**, 109-121 (2017).
8. Pardi, N. et al. Characterization of HIV-1 Nucleoside-Modified mRNA Vaccines in Rabbits and Rhesus Macaques. *Mol Ther Nucleic Acids* **15**, 36-47 (2019).
9. Pardi, N. et al. Expression kinetics of nucleoside-modified mRNA delivered in lipid nanoparticles to mice by various routes. *J Control Release* **217**, 345-351 (2015).
10. Pardi, N. et al. Administration of nucleoside-modified mRNA encoding broadly neutralizing antibody protects humanized mice from HIV-1 challenge. *Nat Commun* **8**, 14630 (2017).

11. Sahin, U., Kariko, K. & Tureci, O. mRNA-based therapeutics--developing a new class of drugs. *Nat Rev Drug Discov* **13**, 759-780 (2014).
12. Jackson, L.A. et al. An mRNA Vaccine against SARS-CoV-2 - Preliminary Report. *N Engl J Med* (2020).
13. Mulligan, M.J. et al. Phase 1/2 study of COVID-19 RNA vaccine BNT162b1 in adults. *Nature* (2020).
14. Laczko, D. et al. A Single Immunization with Nucleoside-Modified mRNA Vaccines Elicits Strong Cellular and Humoral Immune Responses against SARS-CoV-2 in Mice. *Immunity* (2020).
15. Emdin, M. et al. Treatment of cardiac transthyretin amyloidosis: an update. *Eur Heart J* **40**, 3699-3706 (2019).

Reviewers' Comments:

Reviewer #2:

Remarks to the Author:

I still have two concerns.

1. The characterization of modified VEGC-C mRNA-LNC is still not well addressed.
2. The study did not address VEGF-C mRNA as a control.

However, I agree that the current manuscript addressed most of the concerns of the reviewer.

Ms. Ref. No.: NCOMMS-19-01080D

Title: Nucleoside-Modified VEGFC mRNA Induces Organ-Specific Lymphatic Growth and Reverses Experimental Lymphedema

Reviewer #2 (Remarks to the Author):

Reviewer's comment: *I still have two concerns.*

1. The characterization of modified VEGFC mRNA-LNP is still not well addressed.

Murine VEGFC mRNA-LNPs were manufactured and characterized in a rigorous manner. The quality control was ensured during the production process. The mRNA was produced using T7 RNA polymerase (Megascript, Ambion) on linearized plasmids encoding codon-optimized mouse Vascular Endothelial Growth Factor C (pTEV-muVEGFC-A101). mRNAs were transcribed to contain 101 nucleotide-long poly(A) tails. One-methylpseudouridine (m¹Ψ)-5'-triphosphate (TriLink) instead of UTP was used to generate modified nucleoside-containing mRNA. RNAs were capped using the m7G capping kit with 2'-O-methyltransferase (ScriptCap, CellScript) to obtain cap1. mRNA was purified by Fast Protein Liquid Chromatography (FPLC) (Akta Purifier, GE Healthcare), as described. All mRNAs were analyzed by agarose gel electrophoresis and were stored frozen at -20°C. VEGFC-encoding mRNAs were encapsulated in LNPs using a self-assembly process in which an aqueous solution of mRNA at pH=4.0 is rapidly mixed with a solution of lipids dissolved in ethanol. LNPs used in this study were similar in composition to those described previously, which contain an ionizable cationic lipid (proprietary to Acuitas)/phosphatidylcholine/cholesterol/PEG-lipid (50:10:38.5:1.5 mol/mol) and were encapsulated at an RNA to total lipid ratio of ~0.05 (wt/wt). They had a diameter of ~80 nm as measured by dynamic light scattering using a Zetasizer Nano ZS (Malvern Instruments Ltd, Malvern, UK) instrument. mRNA-LNP formulations were stored at -80°C at a concentration of mRNA of 1 µg/µl. All mRNA-LNP formulations for this study were provided after rigorous quality control (size and integrity) to avoid batch-to-batch variations.

All these details have been included in the final manuscript.

Additionally, prior to the *in vivo* experiments, we demonstrated VEGFC protein production from VEGFC mRNA in *in vitro* cell transfection experiments (Suppl. Figs 2a-c).

Taken together, we believe that these detailed physicochemical and *in vitro* analyses are sufficient to characterize VEGFC mRNA-LNPs.

Reviewer's comment: *2. The study did not address VEGF-C mRNA as a control.*

We believe that using naked (unformulated) VEGFC mRNA as a control is not sufficient and the most appropriate control for VEGFC mRNA-LNP administration is the injection of an LNP-formulated irrelevant RNA (poly(C)) that we are using in all of our studies.

The use of LNPs is critical because they facilitate uptake of mRNA and protect it from extracellular RNases. Pardi et al. demonstrated that intradermally administered LNP-formulated mRNA gets translated for a longer time than unformulated (naked) mRNA and more protein is produced from the same amount of LNP-formulated mRNA than from naked mRNA (Figure 1 of Pardi et al, J Exp Med, 2018). These are all critical features for therapeutic activity and justify the use of LNPs. LNPs are critical for durable and high level protein production from mRNA. Pardi et al demonstrated that intradermally administered LNP-formulated mRNA gets translated for a longer period of time than the unformulated (naked) mRNA and a higher amount of protein is produced from the same amount of LNP-formulated mRNA than from naked mRNA. The technology we use is well-established and safe (currently being used in clinical practice).

However, I agree that the current manuscript addressed most of the concerns of the reviewer. We appreciate the feedback from Reviewer #2 and thank the helpful comments and suggestions.